# Graph of States: Solving Abductive Tasks with Large Language Models

Yu Luo[1]   Rongchen Gao[1]   Lu Teng[2]   Xidao Wen[3]   Jiamin Jiang[1]   Qingliang Zhang[1]   Yongqian Sun *[1]
Shenglin Zhang[1]   Jiasong Feng[4]   Tong Liu[4]   Wenjie Zhang[4]   Dan Pei[5]

## Abstract

Logical reasoning encompasses deduction, induction, and abduction. However, while Large Language Models (LLMs) have effectively mastered the former two, abductive reasoning remains significantly underexplored. Existing frameworks, predominantly designed for static deductive tasks, fail to generalize to abductive reasoning due to unstructured state representation and lack of explicit state control. Consequently, they are inevitably prone to *Evidence Fabrication*, *Context Drift*, *Failed Backtracking*, and *Early Stopping*. To bridge this gap, we introduce Graph of States (*GoS*), a general-purpose neuro-symbolic framework tailored for abductive tasks. *GoS* grounds multi-agent collaboration in a structured belief states, utilizing a causal graph to explicitly encode logical dependencies and a state machine to govern the valid transitions of the reasoning process. By dynamically aligning the reasoning focus with these symbolic constraints, our approach transforms aimless, unconstrained exploration into a convergent, directed search. Extensive evaluations on two real-world datasets demonstrate that *GoS* significantly outperforms all baselines, providing a robust solution for complex abductive tasks. Code repo and all prompts: https://github.com/gaorch85/Graph-of-States.

## 1. Introduction

Logical reasoning constitutes the cognitive cornerstone of artificial intelligence, fundamentally categorized into three paradigms: *deduction*, *induction*, and *abduction*. While deduction derives definitive conclusions from general premises and induction generalizes rules from specific observations, abductive reasoning infers the most probable hypotheses from incomplete observations. In the era of LLMs, induction is an inherent capability established through extensive pre-training (Olsson et al., 2022). Meanwhile, deductive tasks (*e.g.*, mathmatics, game of 24) has been substantially advanced by general reasoning frameworks such as Chain-of-Thought (CoT) (Wei et al., 2022) and Tree-of-Thought (ToT) (Yao et al., 2023), with strong performance already reported on representative deductive benchmarks. For example, GPT-5.1 with standard CoT achieves 99.1% on MATH500 (Lightman et al., 2023), 95.7% on AIME2025, and 87.3% on GPQA (Rein et al., 2024). In contrast, the domain of abductive reasoning remains underexplored. Given that abduction serves as the bedrock for decision-making in high-stakes, real-world scenarios (*e.g.*, medical diagnosis, criminal investigation, failure diagnosis in distributed systems), this neglect represents a critical gap. Therefore, we aim to establish a general-purpose reasoning framework tailored for the dynamic and non-monotonic nature of abductive tasks.

As abductive tasks are characterized by incomplete initial information, requiring dynamic evidence investigation to progressively converge the hypothesis space and infer the most plausible cause, directly transposing existing deductive frameworks to these scenarios proves ineffective. To empirically validate this gap, we applied four deductive frameworks (*i.e.*, CoT, ToT, GoT (Besta et al., 2024), FoT (Bi et al., 2025)) to abductive scenarios and conducted a granular analysis of their reasoning trajectories (detailed in Appendix D). As visualized in *Figure 1*, we identified four deficiencies: **(1) Evidence Fabrication:** In an attempt to maintain logical consistency, the model tends to hallucinate non-existent evidence to support biased hypotheses, corrupting the ground truth. **(2) Context Drift** (Dongre et al., 2025): The model tends to forget the current investigation progress during long-horizon tasks (Liu et al., 2024), trapping the agent in redundant loops where it repetitively invokes the same tools or revisits hypotheses that were already falsified by prior evidence. **(3) Failed Backtracking:** Unlike deductive tasks where objective validity criteria (*e.g.*, 7 and 9 cannot make it to 24 through $+, -, \times, \div$) trigger immediate pruning, abductive scenarios present ambiguous intermediate results. Lacking explicit hard constraints, mod-

---

[1]Nankai University [2]Wenzhou Medical University [3]Alibaba Cloud [4]Lenovo [5]Tsinghua University. Correspondence to: Yongqian Sun <sunyongqian@nankai.edu.cn>.

*Proceedings of the 43$^{rd}$ International Conference on Machine Learning*, Seoul, South Korea. PMLR 306, 2026. Copyright 2026 by the author(s).

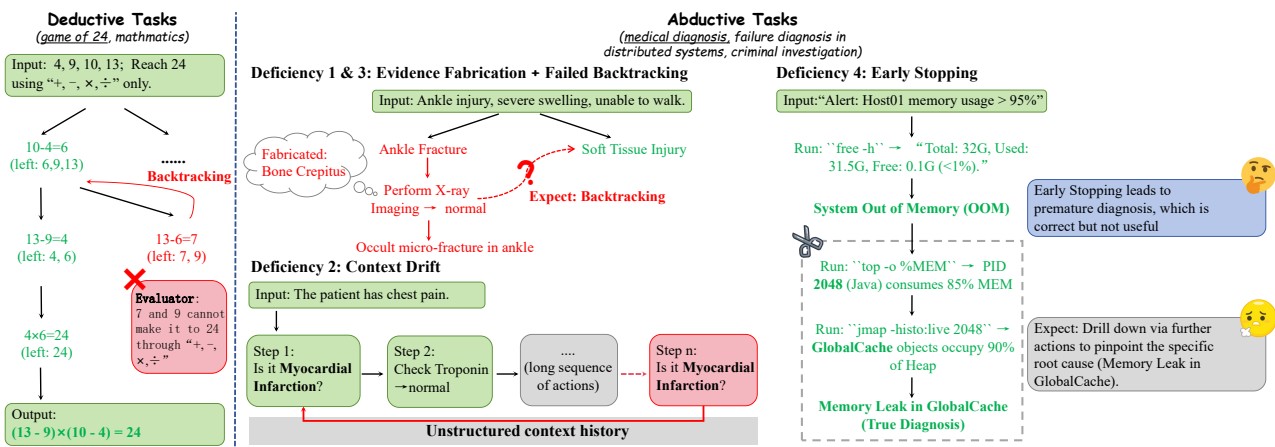

*Figure 1.* Illustration of reasoning frameworks applied to Deductive tasks (Left) versus Abductive tasks (Right). While deductive frameworks succeed in static logic (*e.g.*, Game of 24), applying them to abductive tasks exposes four deficiencies: (1) *Evidence Fabrication*, (2) *Context Drift*, (3) *Failed Backtracking*, and (4) *Early Stopping*.

els often fail to trigger necessary backtracking, persisting on erroneous paths. **(4) Early Stopping:** Models frequently terminate the investigation prematurely upon identifying a superficial symptom, failing to drill down to the fine-grained root cause required for actionable decision-making. We attribute these failures to two fundamental structural limitations in current frameworks. First, the implicit encoding of the reasoning state (*i.e.*, subjective hypotheses and objective facts) within the unstructured context history fails to provide a clear structural representation, directly precipitating *Evidence Fabrication* and *Context Drift*. Second, the lack of state control mechanism relegates backtracking and drill-down decisions to the model's unconstrained autonomy, resulting in *Failed Backtracking* and *Early Stopping*.

While a unified reasoning framework tailored for abductive tasks remains absent, researchers have leveraged LLMs in specialized domains such as medical diagnosis (Kim et al., 2024; Zhou et al., 2025) and failure diagnosis in distributed systems (Zhou et al., 2024; Zhang et al., 2024; Sun et al., 2025; Pei et al., 2025; Luo et al., 2026). To mitigate the aforementioned deficiencies, these approaches incorporate extensive domain-specific adaptations, such as integrating external knowledge bases to curb *Evidence Fabrication* or imposing rule-based heuristics to prevent *Early Stopping*. However, these systems fundamentally rely on conventional deductive reasoning frameworks (*e.g.*, CoT, ToT), attributing their performance gains primarily to heavy domain engineering rather than intrinsic reasoning capabilities. Consequently, such adaptations serve as engineering workarounds that alleviate immediate symptoms while leaving the intrinsic structural deficiencies of the deductive paradigm unaddressed.

To bridge this gap, we introduce Graph of States (*GoS*), a general-purpose reasoning framework tailored for abductive tasks. Our approach employs a dual-layer neuro-symbolic architecture to unify human-aligned collaboration with rigorous logic. In the cognitive layer, we implement a role-based collaborative framework, orchestrating agents aligned with real-world professional roles to ensure a coherent division of labor. Crucially, in the symbolic layer, we introduce two core mechanisms to resolve the aforementioned structural limitations: First, we construct a causal graph as the system's memory, which explicitly structures the belief state by mapping the causal relationships among hypotheses and collected evidence, effectively mitigating *Evidence Fabrication* and *Context Drift*. Second, we employ a state machine as the navigation to govern the reasoning trajectory, enforcing rigorous logical transitions for backtracking and drill-down to prevent *Failed Backtracking* and *Early Stopping*. Quantitative analysis in Appendix D confirms that these mechanisms significantly reduce the frequency of such deficiencies. By dynamically aligning the reasoning focus with these symbolic constraints, *GoS* transforms aimless, unconstrained exploration into a convergent, directed search, ensuring the system steadily approximates the truth.

Our contributions are summarized below: **(1)** We propose *GoS*, and to the best of our knowledge, *GoS* is the first general-purpose multi-agent reasoning framework tailored for abductive reasoning tasks. **(2)** We introduce a neuro-symbolic architecture that leverages causal graph and state machine to construct explicit belief states, thereby transforming unconstrained exploration into directed, convergent search. **(3)** We conduct extensive evaluations on two real-world datasets, demonstrating that *GoS* significantly outperforms existing baselines. **(4)** We make all code and prompts publicly available.

## 2. Related Work

**Reasoning frameworks for general deductive tasks.** A significant body of research has emerged to unlock the potential of LLMs as general problem solvers, though these efforts have predominantly focused on deductive tasks such as the game of 24, mathematical problems, and crossword puzzles. The foundational CoT (Wei et al., 2022) paradigm decomposes complex problems into sequential intermediate steps, where each step forms a coherent natural language sequence contributing to the final solution. Building upon this, subsequent studies have introduced refinements to enhance reliability, including Self-Consistency with CoT (CoT-SC) (Wang et al., 2023), VerifyCoT (Zhao et al., 2023), and Chain of Continuous Thought (Coconut) (Hao et al., 2024). To transcend the limitations of linear reasoning, ToT (Yao et al., 2023) structures the reasoning process as a search over a tree, typically employing algorithms like Depth-First Search (DFS) or Breadth-First Search (BFS) to explore diverse reasoning paths. This non-linear topology is further generalized by frameworks such as Graph of Thought (GoT) (Besta et al., 2024) and Forest of Thought (FoT) (Bi et al., 2025), which extend the structure into graphs and forests to model more complex, non-sequential dependencies.

**Neuro-symbolic reasoning with Large Language Models.** Recent studies have explored combining LLMs with symbolic representations or logic-guided procedures to improve reasoning reliability and faithfulness. For example, LINC (Olausson et al., 2023) translates natural language statements into first-order logic and performs inference with a theorem prover, while LOGIC-LM (Pan et al., 2023) combines symbolic formalization, solver-based reasoning, and self-refinement to improve logical consistency. Aristotle (Xu et al., 2025a) further integrates symbolic logic into decomposition, search, and inference through a logic-complete framework. These methods demonstrate the value of symbolic grounding for improving LLM reasoning. However, they mainly focus on symbolic answer derivation under relatively fixed problem settings, rather than long-horizon interactive investigation that requires iterative evidence acquisition, hypothesis revision, and progressive root-cause refinement.

**Large Language Models for specialized abductive tasks.** Current approaches for specialized abductive tasks mainly employ four categories of domain-specific adaptations designed to mitigate intrinsic reasoning deficiencies and thereby enhance performance: (1) *Multi-Agent Customization*: Frameworks often design specific topological structures to mimic human workflows. For instance, MAM (Zhou et al., 2025) employs a static collaborative topology mimicking a fixed team of specialists to reach consensus, paralleling D-Bot (Zhou et al., 2024), which decomposes the diagnosis process into atomic functional units (*e.g.*, CPU,

Disk) to perform collaborative cross-reviews. (2) *Retrieval-Augmented Generation (RAG)*: To ground reasoning in external knowledge, MDAgents (Kim et al., 2024) augments its diagnostic process with specialized retrieval tools like MedRAG (Xiong et al., 2024). FlowXpert (Shi et al., 2025) employs a hybrid retrieval mechanism querying both vector and graph databases, while Flow-of-Action (Pei et al., 2025) targets the retrieval and matching of high-quality Standard Operating Procedures (SOPs). (3) *Supervised Fine-Tuning (SFT)*: Data-centric approaches prioritize internalizing domain knowledge. Med-PaLM 2 (Singhal et al., 2025) and PMC-LLaMA (Wu et al., 2024) prove that fine-tuning on vast biomedical corpora significantly enhances diagnostic capability, a strategy similarly adopted by LogLM (Liu et al., 2025) using massive log instruction datasets in AIOps. (4) *Data Preprocessing*: Conversely, approaches like TrioXpert (Sun et al., 2025) and OpsAgent (Luo et al., 2026) focus on input abstraction, implementing dedicated pipelines to transform heterogeneous telemetry (*i.e.*, metrics, logs, traces) into structured contexts before feeding them into reasoning modules. To summarize, the efficacy of these approaches is predominantly derived from such **domain-specific engineering** rather than generalized reasoning capabilities.

## 3. Methodology

We propose *GoS*, a novel dual-layer neuro-symbolic framework designed to address complex abductive reasoning tasks by explicitly maintaining reasoning states through causal graphs and state machines (shown in *Figure 2*). Additionally, we introduce a *reasoning focus* to concentrate investigative resources on the most plausible hypothesis, thereby preventing wasteful exploration. This section is organized as follows: Section 3.1 provides the formal definition of the dual-layer framework. Section 3.2 elucidates the bidirectional interaction mechanism between the cognitive and symbolic layers. Finally, Section 3.3 details the state conversions, governing the hierarchical backtracking and drill-down of the inference process.

### 3.1. Dual-layer Framework

**Task Formulation**. We define the abductive reasoning task as an probabilistic inference problem aimed at identifying the most plausible root cause from incomplete information. Let $\mathcal{O} = \{o_1, o_2, ..., o_n\}$ denote the set of observations, which consists of easily accessible surface symptoms $\mathcal{O}_{surf}$ (*e.g.*, chest pain) and costly-to-acquire deep evidence $\mathcal{O}_{deep}$ (*e.g.*, CT scans, ECGs). Given a domain-specific knowledge context $\mathcal{K}$ (*e.g.*, medical guidelines), the objective is to find the optimal $H^*$ from the hypothesis space $\mathcal{H}$ (*e.g.*, myocardial infarction) that maximizes the posterior probability:

$$H^* = \arg\max_{H \in \mathcal{H}} P(H \mid \mathcal{O}_{surf}, \mathcal{O}_{deep}, \mathcal{K}) \qquad (1)$$

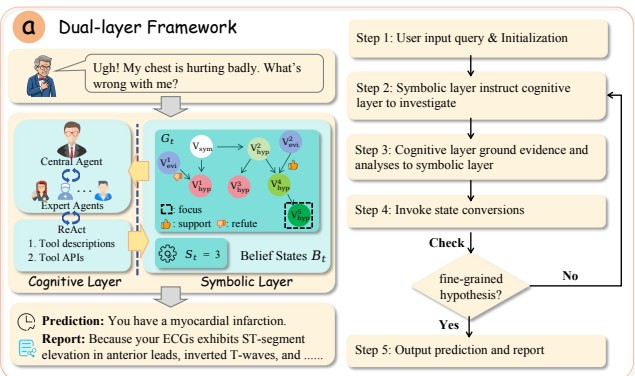

*Figure 2.* **Overview of *GoS*. Left:** Schematic of the dual-layer architecture. **Right:** The iterative inference workflow.

In contrast to passive inference tasks, $\mathcal{O}_{deep}$ is not given as a comprehensive input prior. Instead, it necessitates on-the-fly acquisition, where we employ ReAct (Yao et al., 2022) to execute tool calls for dynamic retrieval.

**Dual-Layer System Definition**. Following are the concrete definition of our system:

1. **The Cognitive Layer ($\mathcal{L}_{cog}$):** This layer functions as the domain-adaptive execution interface of *GoS*, orchestrating a multi-agent system aligned with real-world professional roles to ground the universal reasoning logic into concrete, domain-specific actions. We define the agent system as $\mathcal{A} = \{A_{central}\} \cup A_{experts}$, where one central agent $A_{central}$ acts as orchestrator and several expert agents $A_{experts}$ act as executors. To avoid the reasoning fragmentation caused by atomized functional units, we explicitly design these experts to map coherent real-world professional roles. The rationale is drawn from the division of labor in effective human organizations. Specifically, (1) complex abductive reasoning inherently requires a central orchestrator for global planning and final decision-making, supported by specialized experts for domain-specific execution; and (2) adopting these role divisions, which have persisted through long-term evolution due to their structural robustness, not only enhances collaboration effectiveness but also guarantees a transparent process that is easily understood by humans.

2. **The Symbolic Layer ($\mathcal{L}_{sym}$):** Acting as the *GoS*'s explicit navigational anchor, $\mathcal{L}_{sym}$ formalizes the reasoning status into a structured belief states $\mathcal{B} = (G, S)$, thereby grounding cognitive processes in a transparent and traceable manner distinct from unstructured context history. We formalize the components of the symbolic layer as follows:

**Causal Graph ($G$):** We define a directed graph $G = (V, E)$ to map the logical topology. The node set $V$ comprises three distinct types: symptom ($v_{sym}$), evidence ($v_{evi}$), and hypothesis ($v_{hyp}$), where each hypothesis node is associated with a score $P(v_{hyp})$ representing its confidence. The edge set $E$ encodes three logical primitives: *derive* ($v_{sym} \rightarrow v_{hyp}$) representing initial hypothesis generation, *refine* ($v_{hyp}^{coarse} \rightarrow v_{hyp}^{fine}$) representing granularity evolution of hypothesis, and *support/refute* ($v_{evi} \rightarrow v_{hyp}$) representing evidential confirmation or negation.

**State Machine ($S$):** To capture the hierarchical nature of abductive reasoning, we assign a distinct level $L(v_{hyp})$ to each hypothesis node, representing its semantic granularity. Accordingly, we define a state machine where the state $S_t \in \mathbb{N}^+$ as the level of the hypothesis currently under investigation at step $t$. This state variable governs the overall inference trajectory, facilitating drill-down to finer granularities or backtracking upon contradiction (see Section 3.3).

**The Reasoning Focus ($h^*$):** Formally, at state $S_t$, the reasoning focus $h_t^*$ is defined as the hypothesis with the highest confidence at the current level:

$$h_t^* = \underset{h \in V_{hyp}, \, L(h)=S_t}{\arg\max} P(h \mid G_t) \qquad (2)$$

This formulation enforces a depth-first investigation strategy on the most promising trajectory, thereby facilitating either rapid confirmation or early backtracking to minimize overall investigative costs.

**Initialization**. Upon receiving surface symptoms $\mathcal{O}_{surf}$, the central agent $A_{central}$ constructs the initial graph $G_0$ by deriving preliminary $L(1)$ hypotheses from the root symptom. This establishes the initial belief state $\mathcal{B}_0$ and sets the state machine to $S_0 = 1$, preparing the system for the iterative loop.

**Workflow.** As illustrated in Figure 2, the process begins with user input query and initialization (Step 1). Subsequently, *GoS* enters an iterative reasoning loop: The symbolic layer first identifies the current reasoning focus $h_t^*$ and instructs the cognitive layer to investigate (Step 2). Acting on this directive, the cognitive layer orchestrates agents to gather evidence and updates the causal graph $G_t$ in symbolic layer with new findings (Step 3). Based on the updated graph, the system invokes state conversions (Step 4) to determine the next reasoning depth (*e.g.*, drill down or backtracking). This cycle repeats until a fine-grained hypothesis adequately resolves the input query, producing the final prediction (Step 5).

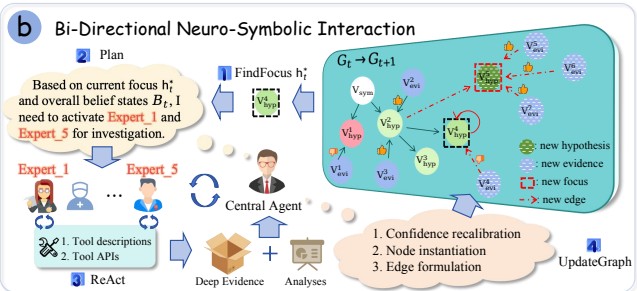

*Figure 3.* **Bi-Directional Neuro-Symbolic Interaction.**

## 3.2. Bi-Directional Neuro-Symbolic Interaction

The core reasoning capability of *GoS* emerges from the bi-directional interaction between the symbolic and cognitive layers. Rather than functioning as isolated modules, these two layers operate within a closed-loop mechanism where explicit state representations and dynamic collaborative reasoning reciprocally inform one another. We detail this bi-directional process through two phases:

**Symbolic-to-Cognitive: Reasoning Focus-Guided Investigation**. In this phase, the symbolic layer transforms the static belief states $\mathcal{B}_t$ into actionable instructions, ensuring that agentic exploration in cognitive layer remains focused and coherent. Unlike XoT frameworks (*e.g.*, CoT, ToT, GoT, FoT), where reasoning states are represented as disordered thought nodes, our symbolic layer explicitly maintains a structured causal graph $G_t$ and identifies a reasoning focus $h_t^*$. This focus represents the most plausible direction of inquiry at the current step, serving as a navigational compass for the cognitive layer. The reasoning focus $h_t^*$ and the belief states $\mathcal{B}_t$ are injected into the central agent $A_{central}$ to formulate executable plans: $Instructions \leftarrow Plan(A_{central}, h_t^*, \mathcal{B}_t)$. These instructions are then dispatched to the corresponding expert agents $A_{experts}$ alongside the global context $(h_t^*, \mathcal{B}_t)$, triggering targeted tool invocation and analysis.

Therefore, anchoring the investigation to focus $h_t^*$ enforces directional stability, effectively mitigating the aimless exploration typical of unstructured reasoning strategies. This focus ensures that computational resources are concentrated on refining the most promising hypothesis. Simultaneously, the shared propagation of $(h_t^*, \mathcal{B}_t)$ establishes a unified cognitive consensus across the multi-agent system. This mechanism prevents expert agents from operating in silos, guaranteeing that distributed reasoning advances coherently without contradicting the global state.

**Cognitive-to-Symbolic: Evidence-Based Grounding**. Following the guidance, expert agents $A_{experts}$ execute ReAct-based tool invocations and analysis. To ground these dis-

tributed investigations, the central agent $A_{central}$ synthesizes the returned observations and analytical results to update the symbolic layer. This aggregation drives the evolution of the causal graph from $G_t$ to $G_{t+1}$ through three key topological operations: (1) *Confidence recalibration*, where the plausibility of existing hypotheses is adjusted by reinforcing confirmed conjectures or diminishing refuted ones based on new evidence; (2) *Node instantiation*, which registers newly discovered evidence and hypotheses into the node set; and (3) *Edge formation*, which establishes the requisite logical dependencies to integrate these new elements into the global reasoning structure. This mechanism ensures that the symbolic layer remains a faithful, real-time reflection of the reasoning progress.

## 3.3. State Conversions: Backtracking and Drill-Down

While the causal graph $G_t$ explicitly maps the structural topology of belief, it acts primarily as a static representation lacking the inherent control logic to drive the progressive deepening of the investigation. Consequently, the state machine $S_t$ is essential to function as the executive controller, regulating the reasoning trajectory through strictly defined transition rules. Drawing inspiration from cognitive models of human abductive reasoning (Elstein et al., 1978; Josephson & Josephson, 1996; Magnani, 2011), which characterize abductive reasoning as an iterative process of hierarchical specification and non-monotonic belief revision, we propose two distinct conversion moves: *backtracking* and *drill-down* (detailed in Algorithm 2).

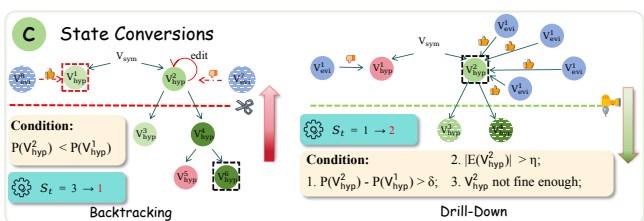

*Figure 4.* **State Conversions: Backtracking & Drill-Down.**

**Backtracking**. The system incorporates a backtracking mechanism to address the non-monotonic nature of abductive reasoning, where initially plausible conjectures are refuted by emerging contradictory evidence. Since the validity of a fine-grained hypothesis strictly relies on the correctness of its precursors, the system continuously monitors the ancestral lineage of the current reasoning focus during confidence recalibration (mentioned in Section 3.2). A regression is triggered if any ancestor node at level $l < S_t$ ceases to be the highest-confidence hypothesis among its siblings due to contradictory evidence. To be specific, the state machine identifies the shallowest level $l^*$ containing the demoted ancestor and executes a pruning operation: all

hypothesis nodes with level $L(v_{hyp}) > l^*$ are discarded, enforcing the principle that inferences founded on a flawed premise are inherently invalid. Consequently, the state machine $S_{t+1}$ is reset to $l^*$, compelling cognitive agents to pivot toward previously dormant alternative branches. This self-correction ensures that the system remains resilient to early-stage misconceptions caused by information scarcity, preventing premature closure on incorrect hypotheses.

**Drill-Down.** Beyond the backward regression, the drill-down transition embodies the system's progressive approximation toward fine-grained ground truths, enabling the transition from high-level conjectures to specific root causes. Following the belief updates in the causal graph, the state machine evaluates whether the current belief states warrants a deeper investigation based on a rigorous *dual-threshold mechanism*, governed by a confidence gap $\delta$ and a minimum support evidence count $\eta$. Specifically, to trigger a state transition, the hypothesis with the highest confidence $v_{hyp}^{(1)}$ must satisfy two simultaneous conditions. First, a *confidence gap* is enforced: the probability difference between the top-ranked hypothesis and the second-ranked must exceed a predefined threshold, denoted as $P(v_{hyp}^{(1)}) - P(v_{hyp}^{(2)}) > \delta$. This criterion ensures that the current direction is unambiguously superior to competing alternatives. Second, reflecting the prudent nature required for high-stakes abductive tasks, we impose an *evidential support constraint* $|E_{sup}(v_{hyp}^{(1)})| \geq \eta$, where $E_{sup}$ denotes the set of supporting evidence nodes linked to the hypothesis. This prevents the system from prematurely narrowing the search space based solely on prior probabilities without sufficient empirical grounding.

Upon satisfying these criteria, the central agent assesses the semantic granularity of $v_{hyp}^{(1)}$. If the hypothesis is sufficiently concrete to resolve the query, the inference terminates, yielding the final prediction accompanied by a comprehensive reasoning report; otherwise, the agent generates the next level of sub-hypotheses to refine $v_{hyp}^{(1)}$, and the state machine increments the reasoning depth $S_{t+1} \leftarrow S_t + 1$. Conversely, if thresholds are not met, the system maintains the current state $S_{t+1} \leftarrow S_t$ to conduct further evidence retrieval in the subsequent iteration.

## 4. Experiments

We empirically validate the proposed *GoS* framework on two distinct abductive reasoning tasks: medical diagnosis and failure diagnosis in distributed systems.

**Datasets.** For medical diagnosis, we utilize DiagnosisArena (Zhu et al., 2025), which curates real-world pathological cases reported in top-tier medical journals (*e.g.*, *Lancet*, *NEJM*, *JAMA*). Specifically, we utilized 150 cases, excluding 12 cases for containing factual errors or logical flaws that make the ground truth unreachable (detailed in Ap-

pendix C.1). For failure diagnosis in distributed systems, we constructed a dataset comprising 150 incidents from a large-scale production microservice systems of a global leading IT company *Lenovo*. Comprehensive descriptions of both datasets are provided in Appendix C.

**Baselines.** Given the absence of established frameworks tailored for general abductive reasoning, we synthesize eight baselines derived from the intersection of two dimensions: *agent architecture* (Single-Agent vs. Multi-Agent) and *reasoning topology* (CoT (Wei et al., 2022) vs. ToT (Yao et al., 2023) vs. GoT (Besta et al., 2024) vs. FoT (Bi et al., 2025)). To ensure a fair comparison, we standardize the atomic reasoning unit across all baselines by adopting the ReAct (Yao et al., 2022) paradigm, wherein each node encapsulates an interleaved triplet of *thought*, *action*, and *observation*. Thereby, this configuration ensures a rigorous evaluation covering the full spectrum from solitary linear reasoning to collaborative hierarchical search strategies. And we use `GPT-5.1-2025-11-13` as the backbone of the agents for all baselines methods and *GoS*.

**Evaluation.** To rigorously assess diagnostic performance, we primarily employ **LLM-as-a-Judge** utilizing a standardized 3-point scale: 2 (*Exact Match*), 1 (*Relevant*), and 0 (*Otherwise*). Accordingly, we report two metrics: *Match* (scoring 2) and *Relevant* (scoring 1 or 2). For evaluation prompts, we utilize the official benchmark prompt for medical diagnosis scenario and construct a parallel prompt for failure diagnosis in distributed systems. However, unlike the deterministic nature of distributed system failures, medical diagnosis entails subtle semantic nuances prone to automated misinterpretation. Consequently, we specifically incorporate **Human-as-a-Judge** for the medical domain to ensure reliability, conducted by a researcher with extensive clinical and academic experience.

**Parameter settings.** We maintain identical hyperparameter configurations across all scenarios to facilitate fair comparison. We impose strict retrieval budgets: each expert agent in *GoS* and the Multi-Agent baselines is limited to 3 retrieval actions, while the Single-Agent baselines are capped at 5. We also set the maximum number of neuro-symbolic interaction iterations in *GoS* to 3.

### 4.1. Medical Diagnosis

Medical diagnosis serves as a quintessential high-stakes abductive task, requiring the inference of diseases from clinical observations through rigorous evidence discovery and cautious reasoning.

**Task Setup.** We reframe the task as a dynamic investigation process: Initially provided only with surface symptoms (*i.e.*, chief complaints and physical examinations), the model must explicitly issue auxiliary examination to re-

trieve corresponding records from an external repository to formulate an accurate diagnosis. In *GoS*, we align the cognitive layer with real-world clinical roles, designating the `Primary_Physician` as the central agent $A_{central}$ and the `Laboratory_Physician`, `Pathologist`, `Radiologist` as expert agents $A_{experts}$. Given that MDAgents (Kim et al., 2024) has already mapped agents to real-world roles, we apply this identical agent configuration to the Multi-Agent baselines to ensure consistency.

*Table 1.* Performance of Medical Diagnosis (%)

| Methods | LLM-as-a-Judge | | Human-as-a-Judge | | $/case |
| | *Match* | *Relevant* | *Match* | *Relevant* | |
|---|---|---|---|---|---|
| *GoS* | **31.88** | **74.64** | **39.86** | **78.99** | 0.12 |
| Single/CoT | 21.01 | 47.83 | 24.64 | 48.55 | **0.03** |
| Single/ToT | 18.84 | 47.10 | 19.57 | 45.65 | 0.08 |
| Single/GoT | 21.01 | 50.00 | 22.46 | 52.90 | 0.07 |
| Single/FoT | 21.74 | 58.70 | 21.01 | 61.59 | 0.32 |
| Multi/CoT | 21.01 | 49.28 | 23.19 | 50.72 | 0.07 |
| Multi/ToT | 20.29 | 50.72 | 23.91 | 53.62 | 0.17 |
| Multi/GoT | 21.74 | 52.17 | 23.91 | 55.07 | 0.15 |
| Multi/FoT | 23.19 | 63.04 | 26.09 | 65.94 | 0.73 |

**Results.** As shown in Table 1, *GoS* significantly outperforms all baseline methods under both LLM-as-a-Judge and Human-as-a-Judge evaluation settings. Specifically, under Human-as-a-Judge evaluation, the *Match* metric of *GoS* reaches 39.86%, and the *Relevant* metric reaches 78.99%. Regarding *reasoning topology*, we observe distinct patterns among baselines. The CoT often achieves higher *Match* metrics than ToT, as ToT tends to deviate due to ineffective node evaluation mechanisms. However, extending the topology from trees (ToT) to graphs (GoT) and forests (FoT) yields performance gains, particularly in the *Relevant* metric. FoT emerges as the strongest baseline, suggesting that a broader search space helps capture comprehensive information. Yet, this comes at a steep price: Multi/FoT incurs the highest cost ($0.73/case), whereas *GoS* achieves superior accuracy at a fraction of the cost ($0.12), validating the efficiency of our directed search. In terms of *agent architecture*, Multi-Agent baselines consistently outperform Single-Agent baselines. This advantage stems from the effective division of labor among multiple expert agents, which enables the model to focus on more information sources and thus provide more comprehensive information support for diagnosis. Additionally, we observe that Human-as-a-Judge consistently yields higher scores than LLM-as-a-Judge. This is because LLM-as-a-Judge prioritizes superficial textual similarity while neglecting deep semantic information, whereas professional researchers can comprehend diverse expressions of the same diagnosis and assign rational scores. We further provide a granular error analysis in Appendix D, dissecting distinct failure modes to shed light on current limitations and future directions.

**Ablation Study.** As presented in Table 2, every component of *GoS* is essential, with their removal leading to performance degradation and higher cost. Initially, removing the reasoning focus $h_t^*$ results in a decline in the *Match* (31.88% $\rightarrow$ 19.57%) alongside a slightly cost increase. This underscores its role in enforcing a depth-first investigation strategy, constraining the agents to scrutinize the most promising hypotheses rather than engaging in stochastic exploration. We additionally compare *GoS* with a simpler structured state management variant that maintains hypotheses, evidence, and their support/refute status in structured text, but without an explicit causal graph. While this organized external state improves over *w/o causal graph* (*Match*: 18.12% vs. 12.32%), it still remains well below full *GoS*. This suggests that the gains are not explained by generic context management alone; rather, the explicit causal graph provides more effective support/refute/refine modeling, enabling more consistent hypothesis updates and better backtracking and drill-down. Furthermore, removing the causal graph or state machine causes a catastrophic drop in performance, with *Match* rates falling to 12.32%. This validates that constructing and maintaining belief states are indispensable in complex abductive reasoning tasks. Notably, we observe a cost spike when removing the state machine ($0.17/case). We attribute this increase to the absence of valid state transition protocols, as without a mechanism to explicitly manage the reasoning lifecycle and termination, the system tends to exhaust the full reasoning budget before forcing a conclusion, thereby incurring unnecessary computational overhead.

*Table 2.* Ablation Study of Medical Diagnosis (%)

| Methods | LLM-as-a-Judge | | $/case |
| | *Match* | *Relevant* | |
|---|---|---|---|
| *GoS* | **31.88** | **74.64** | **0.12** |
| w/o reasoning focus | 19.57 | 67.39 | 0.14 |
| w/ structured state management | 18.12 | 59.42 | 0.15 |
| w/o causal graph | 12.32 | 48.55 | **0.12** |
| w/o state machine | 12.32 | 50.00 | 0.17 |

**Sensitivity Analysis.** Figure 5 presents a sensitivity analysis of *GoS* under varying configurations. Regarding reasoning budget (Upper-Left/Right), increasing both interaction iterations and retrieval steps generally enhances performance. Notably, comparisons with the best-performing baseline (dashed line) reveal that *GoS* achieves superior efficiency, surpassing the baseline's peak performance even with a restricted budget. Regarding the dual-thresholds $(\eta, \delta)$ introduced in Section 3.3 (Lower-Left/Right), we observe a distinct trade-off between precision and conservatism. Raising these thresholds imposes stricter criteria for drill-down transitions, compelling the model to accumulate more supporting evidence and ensure higher confidence in the top-ranked hypothesis. While moderate thresholds improve correctness by curbing hasty reasoning, excessively high

thresholds force the model to adopt a conservative strategy: it tends to terminate at superficial diagnosis (boosting *Relevant*) rather than risking a fine-grained root cause prediction without overwhelming evidence (lowering *Match*). This behavior validates that these thresholds act as effective control knobs, allowing users to tune the system's risk tolerance based on deployment needs.

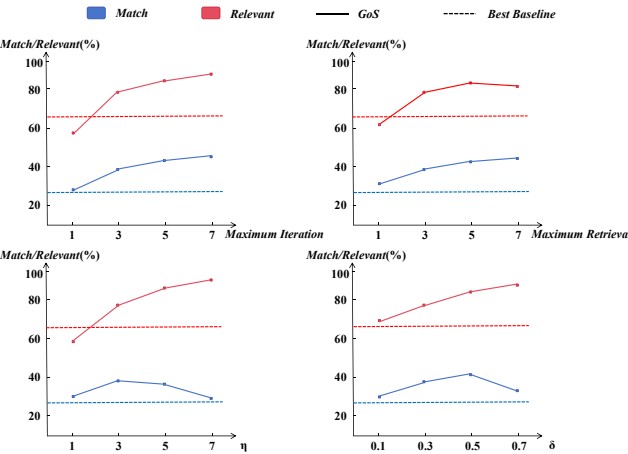

*Figure 5.* Sensitivity Analysis. Solid line stands for *GoS*, dashed line stands for best baseline (Multi/FoT). (1) **upper-left**: maximum number of neuro-symbolic interaction iterations; (2) **upper-right**: maximum number of retrieval actions of expert agent; (3) **lower-left**: minimum support evidence for drill-down transition; (4) **lower-right**: confidence gap for drill-down transition.

## 4.2. Failure Diagnosis in Distributed Systems

Failure diagnosis in distributed systems is a critical operational task to mitigate substantial economic losses, requiring the joint analysis of heterogeneous observability data and system commands to pinpoint root causes.

**Task Setup.** For each case, the model is provided with the initial alert, specifying the alert type, affected components, timestamps, and overall failure descriptions. Crucially, detailed diagnostic evidence is hidden within the massive raw observability data, requiring the model to actively investigate to reconstruct the failure context. To execute this investigation, *GoS* instantiates an `ApplicationOperator` as the central agent ($A_{central}$) orchestrating three expert agents ($A_{experts}$): `LinuxOperator`, `NetworkOperator`, and `DatabaseOperator`. These experts encapsulate specialized capabilities, such as querying historical logs, retrieving system metrics, and executing restricted shell commands. For Multi-Agent baselines, we adopt the domain-oriented design of D-Bot (Zhou et al., 2024), instantiating resource-centric experts (*e.g.*, CPU, Memory, and Disk experts), each with access limited to their respective domain-relevant observability data.

*Table 3.* Performance of Failure Diagnosis in Distributed Systems (%)

| Methods | LLM-as-a-Judge | | $/case |
| --- | --- | --- | --- |
| | *Match* | *Relevant* | |
| *GoS* | **70.67** | **88.00** | 0.10 |
| Single/CoT | 26.67 | 81.33 | **0.03** |
| Single/ToT | 25.33 | 78.00 | 0.14 |
| Single/GoT | 27.33 | 80.00 | 0.11 |
| Single/FoT | 28.67 | 84.00 | 0.45 |
| Multi/CoT | 34.00 | 82.67 | 0.05 |
| Multi/ToT | 25.33 | 81.33 | 0.13 |
| Multi/GoT | 28.00 | 80.67 | 0.18 |
| Multi/FoT | 28.00 | 86.67 | 0.94 |

**Results.** As detailed in Table 3, *GoS* establishes a new SOTA under the LLM-as-a-Judge setting. Specifically, *GoS* achieves 70.67% in *Match* and 88.00% in *Relevant*, surpassing the respective best-performing baselines by margins of 36.67% and 1.33%. We observe that the performance trends regarding *reasoning topology* and *agent architectures* largely mirror those in the medical diagnosis task. Notably, among baselines, Multi/CoT yields the highest *Match* score (34.00%), whereas Multi/FoT attains the highest *Relevant* score (86.67%) but incurs the highest cost ($0.94/case). Despite the competitive *Relevant* scores across baselines, a distinct pattern emerges: they significantly underperform in *Match* compared to *GoS*. We attribute this to the structured nature of system alerts, which contain explicit metadata that easily guides models to the correct failure domain (high *Relevant*). Yet, distinguishing the precise root cause from superficial symptoms requires in-depth investigation. The superior *Match* performance of *GoS* stems from its state machine, which enforces a coarse-to-fine reasoning process. This mechanism effectively refines coarse hypotheses into fine-grained root causes, whereas baselines often stagnate at superficial diagnoses (low *Match*). Regarding cost, *GoS* delivers superior accuracy at a fraction of Multi/FoT's expense ($0.10 vs. $0.94, $\sim$ 8x cost reduction), validating the economic efficiency of *GoS*.

**Example Case.** *Figure 6* presents a real incident from a production host where multiple alerts are triggered. Under the *GoS* framework, the `ApplicationOperator` first aggregates these alerts to initialize several hypotheses and selects "Filesystem Read-Only" as the reasoning focus due to its highest confidence. Then it coordinates the investigation by instructing the `LinuxOperator` to inspect the filesystem status. Through restricted shell inspections, including mount state verification and XFS-related kernel log analysis, the `LinuxOperator` observes that the root filesystem (XFS on dm-0) has been remounted in read-only mode, with kernel logs explicitly reporting XFS metadata corruption. Based on this evidence, *GoS* updates the causal graph by increasing the confidence of the current reasoning

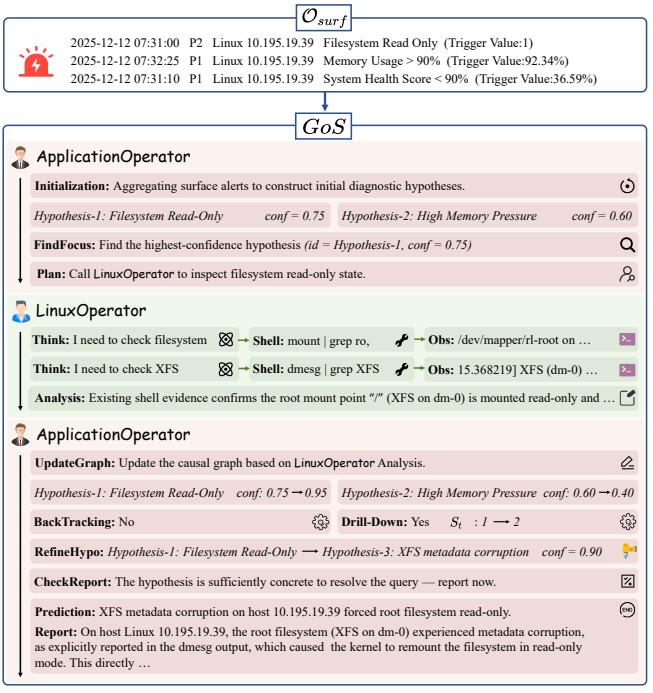

*Figure 6.* An example of failure diagnosis in distributed systems

focus while suppressing competing alternatives, and enables a drill-down transition to a finer-grained root cause. The system finally reports XFS metadata corruption as the root cause, which remounted the root filesystem in read-only mode.

## 5. Conclusion

In this work, we presented *GoS*, a general-purpose neuro-symbolic framework tailored for abductive reasoning. Addressing the structural limitations where deductive paradigms fail, we introduced a dual-layer architecture that grounds collaborative investigation in explicit belief states via a causal graph and a state machine. This design transforms aimless, unconstrained exploration into a directed, convergent search, effectively navigating incomplete information to identify fine-grained root causes. Looking forward, we envision *GoS* serving as a robust reasoning backbone that, when coupled with domain-specific adaptations, paves the way for reliable decision-making across complex, high-stakes real-world domains.

## Acknowledgement

This work is supported by the CCF-Lenovo Blue Ocean Research Fund, the National Natural Science Foundation of China (62272249, 62302244), the Tianjin Key Research and Development Program (Grant No. 25YFYFFG00690), and the Fundamental Research Funds for the Central Universities (XXX-63253249).

## Impact Statement

This paper introduces *GoS*, a dual-layer neuro-symbolic framework that fills a critical blank regarding the absence of general reasoning framework for abductive tasks. By orchestrating a synergy where the symbolic layer maintains explicit reasoning states to guide the cognitive layer's dynamic investigation and execution, our work offers theoretical and practical guidance for complex real-world abductive scenarios such as medical diagnosis, failure diagnosis in distributed systems, and criminal investigation. Importantly, our framework does not negate existing efforts in specialized domains but rather seeks to upgrade the underlying reasoning paradigm. Established domain-specific adaptations, including RAG and input preprocessing, can be seamlessly integrated into *GoS* to further enhance their effectiveness. Given that reliable abductive reasoning is fundamental to decision-making across diverse high-stakes environments, a generalizable framework that enhances the robustness of such systems holds significant potential for positive societal impact.

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

## A. Algorithm

Here we present the pesudo code of the overall workflow of *GoS* in Algorithm 1, and the state conversion mechanism in Algorithm 2.

---

**Algorithm 1** Graph of States

---

1: **Input:** surface symptoms $\mathcal{O}_{surf}$, central agent $A_{central}$, expert agents $A_{experts}$
2: $ReportFlag = false$
3: $B_0, G_0, S_0$ = Initialization($A_{central}, \mathcal{O}_{surf}$)
4: **repeat**
5:     // Symbolic-to-Cognitive
6:     $h_t^* = $ FindFocus($G_t$)
7:     $Instructions = $ Plan($A_{central}, h_t^*, \mathcal{B}_t$)
8:     $\mathcal{O}_{deep}, analyses = $ ReAct($A_{experts}, Instructions, h_t^*, \mathcal{B}_t$)
9:     // Cognitive-to-Symbolic
10:     $G_{t+1} = $ UpdateGraph($A_{central}, analyses, G_t$)
11:     // State Conversions, detailed in Algorithm 2
12:     $S_{t+1}, G_{t+1}^* = $ StateConversion($A_{central}, G_{t+1}, S_t$)
13:     $ReportFlag = $ CheckReport($G_{t+1}^*$)
14: **until** $ReportFlag$ is $true$
15: prediction, report = Report($A_{central}, \mathcal{B}_{t+1}$)
16: **Return:** prediction, report

---

**Algorithm 2** State Conversions

---

1: **Input:** graph $G_{t+1}$, gap_delta $\delta$, min_support $\eta$
2: // Backtracking
3: $BacktrackFlag = false$
4: $BacktrackFlag, l^* = $ CheckBacktrack($G_{t+1}$)
5: **if** $BacktrackFlag$ is $true$ **then**
6:     **for** $L(v_{hyp}^i) > l^*$ **do**
7:         Delete($v_{hyp}^i$)
8:     **end for**
9: **end if**
10: // Drill-down
11: **if** $P(v_{hyp}^{(1)}) - P(v_{hyp}^{(2)}) > \delta$ and $|E_{sup}(v_{hyp}^{(1)})| \geq \eta$ **then**
12:     // No drill-down if granularity is fine enough
13:     **if** CheckGranularity($A_{central}, G_{t+1}$) is $true$ **then**
14:         **Return:** $S_t, G_{t+1}$
15:     **end if**
16:     $S_{t+1} = S_t + 1$
17:     // Generate fine-grained hypothesis
18:     $G_{t+1}^* = $ RefineHypo($A_{central}, G_{t+1}$)
19: **else**
20:     $S_{t+1} = S_t$
21: **end if**
22: **Return:** $S_{t+1}, G_{t+1}^*$

---

## B. Limitations

While our experiments demonstrate the efficacy of *GoS* across diverse domains, we acknowledge certain limitations regarding the breadth of our evaluation. Currently, our experimental scope is confined to medical diagnosis and failure diagnosis in distributed systems. Although abductive reasoning is critical in other high-stakes fields such as criminal investigation, the

acquisition of high-quality benchmarks in these domains remains a significant challenge due to strict privacy regulations and data protection laws, which limit access to the detailed evidential data required to construct authentic abductive scenarios. Nevertheless, the datasets employed in this study are of exceptional quality—sourced from top-tier medical journals and real-world production environments of leading technology companies—thereby ensuring that our findings regarding the model's reasoning capabilities remain robust and representative (see Appendix C for more detail).

## C. Descriptions of datasets

### C.1. Dataset for Medical Diagnosis

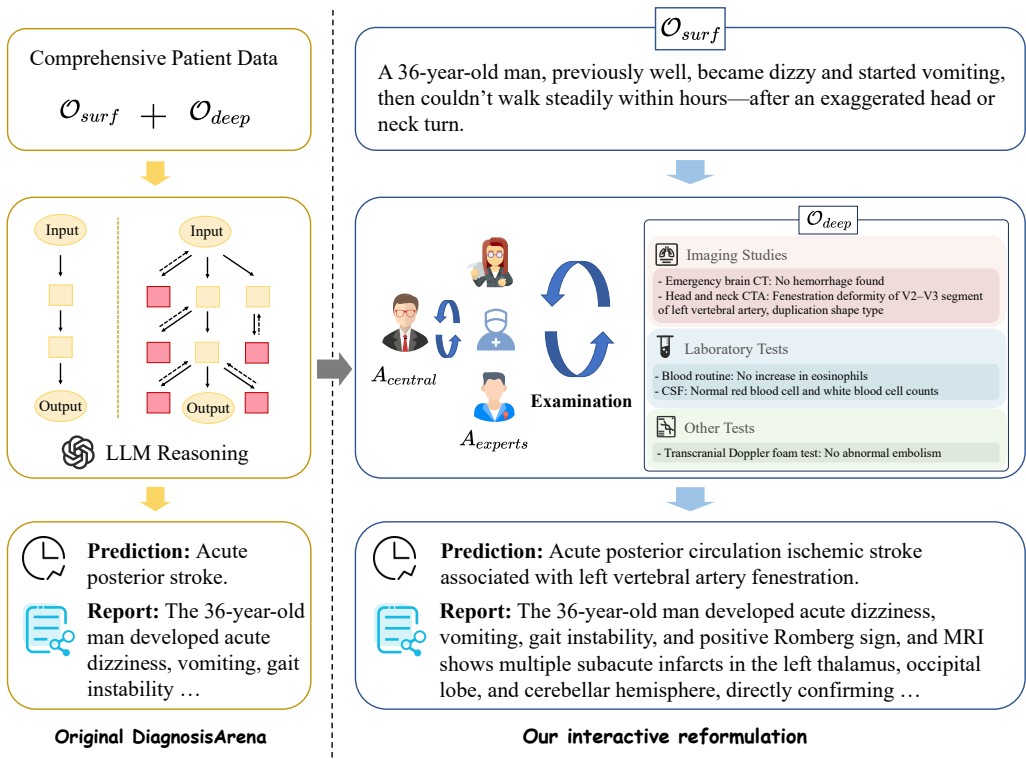

*Figure 7.* Reformulating DiagnosisArena as an interactive diagnostic task. In the original setting, all auxiliary examination records are revealed upfront. In contrast, our reformulation restricts the initial input to the chief complaint and basic physical examination, and requires the model to explicitly request auxiliary examinations from an external repository before making the final diagnosis.

**Why do we use DiagnosisArena?** We utilize DiagnosisArena (Zhu et al., 2025), a benchmark curated from case reports published in top-tier medical journals (*e.g.*, Lancet, NEJM, JAMA) to evaluate diagnostic reasoning capability. Benchmarks derived from medical licensing examinations, such as MedQA (Jin et al., 2021) and MedMCQA (Pal et al., 2022), primarily focus on evaluating the retention of standardized medical knowledge rather than the reasoning capabilities required for diagnosis. In contrast, DiagnosisArena simulates complex abductive reasoning scenarios over multidimensional patient records, and it is subjected to rigorous human verification to guarantee high data quality and clinical fidelity.

**How do we reframe the task?** As illustrated in Figure 7, the original DiagnosisArena benchmark presents models with comprehensive patient data upfront, including all critical auxiliary examination records (*e.g.*, laboratory tests and imaging) alongside the initial symptoms. This setting significantly trivializes the diagnostic challenge, as in real-world clinical practice, auxiliary examinations incur substantial temporal and financial costs. Consequently, the core difficulty lies in identifying the necessary examinations to accumulate evidence, which is eliminated in its original setting. To simulate this realistic constraint, we reframe the task as a dynamic investigation process. We restrict the initial input strictly to the patient's chief complaint and basic physical examination, encapsulating all auxiliary examination results within an external information repository. The model must explicitly issue precise auxiliary examination commands to retrieve corresponding records; otherwise, no information is revealed. This formulation effectively restores the inference difficulty and faithfully

mirrors the active, evidence-seeking nature of medical diagnosis.

| | LLM-as-a-Judge | | | | |
|---|---|---|---|---|---|
| | I | II | III | IV | V |
| Case 1 | 1 | 1 | 1 | 1 | 1 |
| Case 2 | 0 | 0 | 0 | 0 | 0 |
| Case 3 | 0 | 1 | 1 | 1 | 1 |
| Case 4 | 2 | 2 | 2 | 2 | 2 |
| Case 5 | 1 | 1 | 1 | 1 | 1 |
| Case 6 | 2 | 2 | 1 | 2 | 2 |
| Case 7 | 0 | 1 | 1 | 0 | 0 |
| Case 8 | 1 | 1 | 1 | 1 | 1 |
| Case 9 | 0 | 0 | 0 | 0 | 0 |
| Case 10 | 2 | 1 | 2 | 1 | 0 |

*Figure 8.* Scoring consistency of LLM-as-a-Judge across five repetitions for ten medical diagnostic cases (green: 6 consistent; red: 4 inconsistent).

**Why do we incorporate Human-as-a-Judge?** Although LLM-as-a-Judge demonstrates strong alignment with human evaluations in general domains due to large-scale Reinforcement Learning from Human Feedback (RLHF), its instability in medical diagnosis necessitates human oversight. Empirical analysis reveals that identical diagnostic outputs receive inconsistent scores across repeated LLM assessments (as shown in *Figure* 8), with unresolved ambiguities persisting even after majority voting. Crucially, LLMs exhibit a pronounced bias toward longer responses, prioritizing length over diagnostic accuracy—a direct consequence of the high domain specificity and insufficient medical expertise in current language models. To ensure robust evaluation, we introduce Human-as-a-Judge: a clinical researcher with extensive academic and clinical experience performed **110+** hours of manual assessment, strictly prioritizing diagnostic correctness against ground truth. This approach enables granular error analysis, uncovering systematic model failures that directly inform targeted model refinement and future clinical deployment.

**Why do we need to remove several cases?** Following manual verification, we excluded 12 cases containing critical information gaps or logical contradictions that render the labeled diagnosis impossible to derive from the provided evidence. To substantiate the necessity of this filtration and strictly maintain the integrity of our benchmark, we provide a detailed analysis of three representative examples below. These instances illustrate specifically how such intrinsic defects obstruct the valid diagnostic reasoning process.

---

**Case#1: Labeled answer is a secondary diagnosis unrelated to the chief complaint**

**Brief description:** A man in his 90s with a history of hypertension, hypercholesterolemia, sinus node dysfunction, and prior dual-chamber pacemaker implantation presented with 1 month of abdominal pain radiating to his back.
**Labeled Answer:** Pacemaker pseudofusion.
**Why:** In this case, the patient presented primarily with abdominal pain radiating to the back for 1 month. Auxiliary examinations identified a highly likely cause of the abdominal pain: a $5.3 \times 5.3$-cm infrarenal abdominal aortic aneurysm with thrombus. Both the chief complaint and auxiliary examinations indicate that the patient's abdominal pain is caused by the abdominal aortic aneurysm, which is highly consistent with the answer generated by *GoS*. The labeled answer provided by the dataset is "Pacemaker pseudofusion", which may be inferred from the absence of atrial or ventricular pacing signals on the electrocardiogram. However, this diagnosis cannot explain the patient's abdominal pain radiating to the back and thus cannot serve as the primary diagnosis. Therefore, the reason for excluding this case is that the labeled answer cannot serve as the patient's primary diagnosis but only as a secondary

one, which deviates from the core issue addressed by the chief complaint.

---

**Case#2: Incorrect labeled answer**

**Brief description:** A 36-year-old woman (gravida 2, para 2) presented with right lower pelvic pain. The patient had been previously diagnosed with presumed progressive uterine fibroids 6 months earlier and sought a second opinion.
**Labeled Answer:** Tubal angiomyofibroblastoma
**Why:** The reason for excluding this case is that the labeled answer provided by the dataset is **incorrect**. The answer given is "Tubal angiomyofibroblastoma", while we confirm the diagnosis as uterine leiomyoma. Firstly, based on the chief complaint and imaging findings, this case can be identified as a tumor-related disease; therefore, the gold standard for diagnosis should be pathological examination. Subsequently, we focused on the information from the pathological examination and presented three pieces of evidence inconsistent with "Tubal angiomyofibroblastoma":

1. Mismatched immunohistochemical (IHC) profile: The pathological examination demonstrated "Desmin (desmin) positive + ER/PR positive + CD34 negative" — whereas the typical IHC features of angiomyofibroblastoma are "Desmin negative/weakly positive + CD34 positive + ER/PR mostly negative". Immunohistochemistry serves as the "identity code" for tumor differentiation. The reverse expression of Desmin (a smooth muscle-specific marker, positive) and CD34 (positive in angiomyofibroblastoma) constitutes the core differential basis between the two entities. A discrepancy in a single key indicator is sufficient for exclusion.

2. Mismatched morphological characteristics: The pathological examination revealed "spindle cells and cords of epithelioid cells surrounding numerous blood vessels + loose fibromyxoid stroma + well-demarcated border" — while the typical morphological hallmarks of angiomyofibroblastoma include "angiocentric clustered/nested arrangement (cells densely surrounding blood vessels to form nodules) + alternating dense and sparse cellular areas + stroma may be myxoid but not the typical loose fibromyxoid type". Although both tumors exhibit "perivascular growth", the "angiocentric clustering" of angiomyofibroblastoma is a signature feature. The report only describes cellular cords around blood vessels without dense clustering/nesting, and the stroma is typically loose fibromyxoid, which is inconsistent with the structural characteristics of angiomyofibroblastoma.

3. Hormone correlation: The report indicates ER/PR positivity, suggesting the tumor is hormonally regulated.

   In contrast, angiomyofibroblastoma has no association with hormones (both ER and PR are negative), further supporting the exclusion of this diagnosis.

---

**Case#3: Insufficient clinical data to derive the labeled answer**

**Brief description:** A 57-year-old male patient was admitted with breathlessness, cough and weakness from the day before. He had a history of a head injury 20 years ago which resulted in cerebral atrophy, quadriplegia and aphasia. The patient had a history of pulmonary thromboembolism in the past year which was under treatment. He also had a history of hypertension and several hospitalizations.
**Labeled Answer:** NDM-positive Burkholderia cepacia complex (Bcc) lower respiratory tract infection
**Why:** The main reason for excluding this case is the insufficient information provided, which is inadequate to support a complete answer and thereby affects the accuracy of the response generated by the model. The labeled answer specifies the multidrug-resistant (MDR) bacterium as Burkholderia cepacia complex (Bcc) — a result that requires specific bacterial species identification to confirm. However, the available data only supports the diagnosis of "multidrug-resistant Gram-negative bacteria," so *GoS* solely provided the answer: "MDR Gram-negative bacterial aspiration pneumonia causing acute lower respiratory infection."

---

## C.2. Dataset for Failure Diagnosis in Distributed Systems

**Why do we construct a real-world distributed systems incident dataset?** Existing RCA/AIOps benchmarks have enabled important progress, but the prevailing evaluation setups still differ from live incident response in two key ways: (1) some benchmarks are evaluated under synthetic or limited-scale settings, which may not reflect the complexity and heterogeneity of production incidents (Li et al., 2022a; Ikram et al., 2022); and (2) many benchmarks define diagnosis

around a narrow target (*e.g.*, predicting only one type of root-cause element), which can encourage goal-specific solutions and limit transferability across incident objectives (Li et al., 2022b; Lee et al., 2023; Yu et al., 2023). OpenRCA (Xu et al., 2025b) takes a meaningful step toward realism by providing multi-source telemetry from production systems and adopting a goal-driven task formulation, yet it still largely treats diagnosis as an offline problem where evidence is assumed to be pre-materialized at inference time (and, when first-hand incident reports are unavailable, queries are approximated via synthesis). In contrast, real-world failure diagnosis is inherently interactive under partial and evolving observability: on-call engineers iteratively probe the system by issuing shell-level commands, inspecting live states, and acquiring additional signals on demand (Figure 9), rather than passively analyzing a fixed telemetry corpus. To faithfully evaluate diagnostic reasoning in this evidence-seeking setting, we construct a dataset directly from authentic production incidents, explicitly capturing intermediate shell snapshots and on-demand observations along the investigation trajectory.

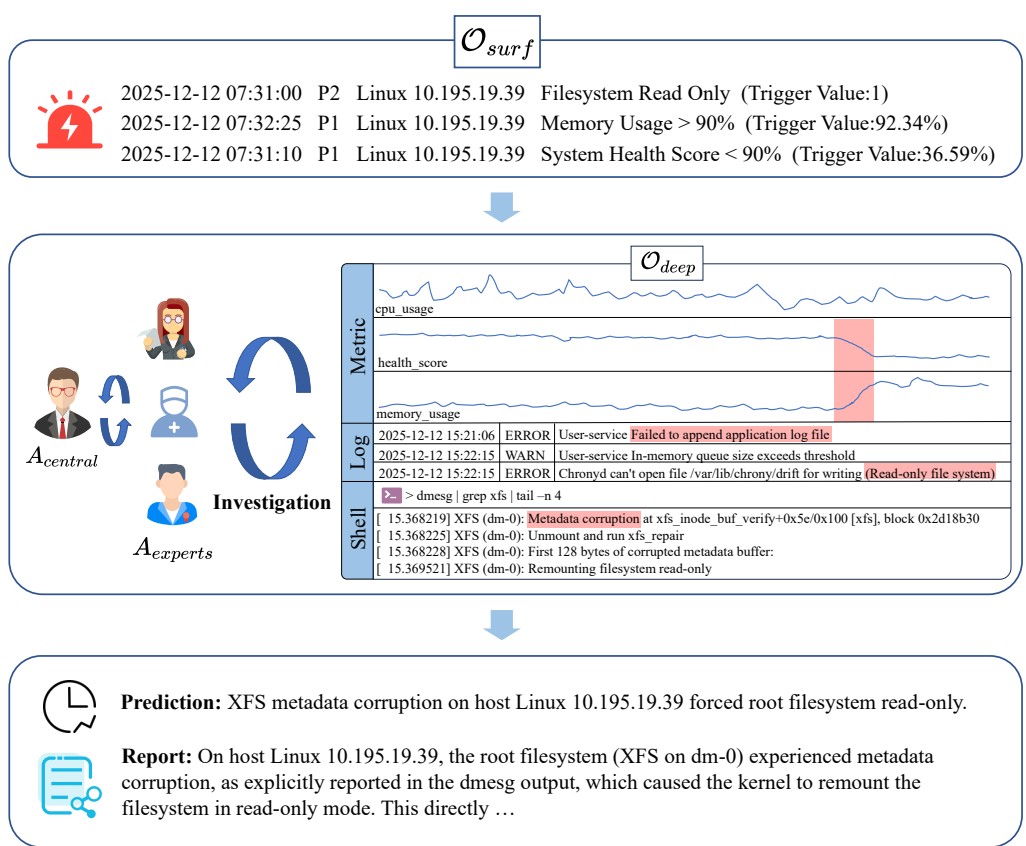

*Figure 9.* An example of an interactive failure diagnosis process in distributed systems. Given an initial alarm summary, the agent iteratively investigates the incident by querying heterogeneous signals, including time-series metrics, system logs, and shell-level snapshots. Through progressive evidence collection and interpretation, the system converges to the final root cause.

**Dataset characteristics.** We collect 150 production incidents and represent each as an incident-centric investigation trajectory, spanning from the first alert to the postmortem-confirmed root cause. Each incident records heterogeneous diagnostic signals observed during live response, including time-series metrics, system logs, and shell-level snapshots. The root cause labels are derived from post-incident analyses conducted by experienced on-call engineers. Importantly, shell snapshots capture intermediate probing actions and raw system states (*i.e.*, command outputs) during troubleshooting, rather than curated summaries or oracle evidence. Consequently, the root cause is not explicitly revealed unless the collected evidence is correctly interpreted and integrated over the investigation over time.

**Scale and coverage.** The incidents are drawn from an industrial observability platform monitoring on the order of $2.5 \times 10^4$ infrastructure instances, spanning multiple layers of modern stacks: data-center facilities (*e.g.*, temperature/humidity sensors, precision cooling, backup power), infrastructure hardware (mainstream server families), virtualization and cloud (*e.g.*, VMware/OpenStack and major public clouds), hosts (*e.g.*, Linux/Windows/AIX), middleware (*e.g.*, Kafka/RabbitMQ), databases (*e.g.*, PostgreSQL/MySQL/MongoDB/Redis), and application/service-level end-to-end monitoring (including

enterprise analytics workloads such as HANA). The platform ingests network log streams at the order of $10^{10}$ entries per day (*e.g.*, ~12B/day) and retains PB-scale historical observability data, which makes root-cause localization depend on efficient, targeted evidence seeking rather than exhaustive log inspection.

**Data availability.** The incident dataset used in this work is collected from a real-world production distributed system and contains sensitive operational information (*e.g.*, logs/metrics/traces) and proprietary identifiers. Due to company confidentiality, privacy, and compliance requirements, we cannot publicly release the raw data or interaction traces. All case examples shown in this paper are sanitized and anonymized, and do not correspond to real asset identities.

## D. Error Analysis

To gain a deeper understanding of the limitations of current neuro-symbolic approaches and the inherent challenges of abductive reasoning tasks. We conducted a manual inspection over failure cases (score smaller than 2) from both *GoS* and all baseline methods across the domains of medical diagnosis and failure diagnosis in distributed systems. This analysis identifies common pitfalls and offers guidance for future architecture optimization.

### D.1. Taxonomy of Error Types

To systematically dissect the limitations of *GoS* and understand the underlying causes of diagnostic failures, we synthesized the observed failures into five primary error types based on empirical frequency. Note that this taxonomy highlights the most prevalent error types rather than being exhaustive, with negligible outliers omitted. Furthermore, these categories are not mutually exclusive, as a single failure case may exhibit compound errors spanning multiple types. This taxonomy ranges from low-level operational missteps to high-level strategic deficiencies in abductive reasoning. The definitions and representative examples for each error type are detailed below:

1. **Wrong Action Selection.**
   This error occurs at the operational level when the agent invokes tools that are logically irrelevant to the current hypothesis. It reflects a disconnect between the reasoning intent and the executed action, resulting in resource wastage without yielding valid evidence.
   *Example: The model suspects a CPU saturation issue but erroneously invokes tools to retrieve memory-related metrics, failing to verify the initial suspicion.*

2. **Evidence Fabrication.**
   This refers to the hallucination phenomenon where the model generates non-existent evidence to support a biased hypothesis. Unlike logical errors, this involves the corruption of the ground truth, where the model explicitly asserts the presence of specific observational details that are entirely absent from the provided context.
   *Example: To support a "Database Connection Failure" hypothesis, the model claims to find a specific "Connection Refused" error entry in the application logs, even though the retrieved log file is actually normal.*

3. **Context Drift.**
   This error manifests as a failure in long-term state maintenance. The model loses track of the historical reasoning trajectory, leading to the redundant execution of previously used tools or the resurrection of hypotheses that were already falsified in earlier turns.
   *Example: The agent queries the database latency again in Turn 10, ignoring that the same query in Turn 3 had already ruled out database issues.*

4. **Failed Backtracking.**
   This represents a critical strategic flaw in non-monotonic reasoning. When encountering evidence that refutes the current deduction, the model fails to revert to a prior state to explore alternative branches. Instead, it exhibits "logical stubbornness" by proposing ad-hoc, low-probability auxiliary hypotheses to force compatibility with the contradictory evidence.
   *Example: When an X-ray result rules out an "ankle fracture," instead of pivoting to soft tissue damage, the model doubles down with "occult micro-fracture," a rare condition undetectable by X-ray, to defend its initial guess.*

5. **Early Stopping.**
   This error occurs when the reasoning process halts at a coarse-grained symptom level rather than drilling down to the

root cause. While the diagnosis is directionally correct (yielding high *Relevant* scores), it lacks the granularity required for actionable mitigation (resulting in low *Match* scores).

***Example:*** *The model correctly identifies a "Memory Fault" and terminates, failing to investigate further to pinpoint the specific "Memory Exhaustion" caused by a memory leak, which is necessary to trigger the correct scaling action.*

### D.2. Quantitative Distribution Analysis

To rigorously quantify the failure mechanisms, we conducted a statistical analysis over failure cases, dissecting the error distributions from two distinct perspectives: (1) a methodological comparison between *GoS* and the aggregated baselines, where error prevalence is calculated over the pooled failure cases of all baseline models across both scenarios, and (2) a cross-domain comparison analyzing failure cases exclusively within *GoS*. It is important to note that these error types are neither mutually exclusive nor do they cover every potential error type. A single reasoning trajectory may exhibit compound failures (*e.g.*, an "*Evidence Fabrication*" leading to a subsequent "*Failed Backtracking*"), and minor anomalies outside our primary taxonomy are not included. Consequently, the reported statistics reflect the prevalence of each error type relative to the total number of failure cases, and the cumulative percentages may not strictly sum to 100%. The detailed distributions are presented in Table 4 and Table 5.

*Table 4.* Error Distribution Comparison across Methods (%).

| Methods | Error Types | | | | |
|---|---|---|---|---|---|
| | Wrong Action Selection | Evidence Fabrication | Context Drift | Failed Backtracking | Early Stopping |
| Baselines (Pooled) | 47.22 | 22.22 | 41.32 | 52.78 | **63.89** |
| *GoS* | **55.90** | 0 | 9.70 | 30.90 | 18.75 |

**Methodological Comparison.** Table 4 reveals a fundamental shift in error dynamics between *GoS* and baselines. We analyze these variations through two distinct lenses: architectural constraints and domain knowledge dependency.

1. **Architectural Advantages.** The most significant improvement lies in the elimination of *Evidence Fabrication* in *GoS* (0% vs. 22.22%). This is a direct consequence of our neuro-symbolic architecture, which imposes strict grounding constraints: the construction of nodes and edges in the causal graph must be derived from actual tool execution returns. Unlike baselines, which generate unconstrained text and are prone to hallucinating non-existent evidence, *GoS* cannot "invent" evidence without interaction. Similarly, the explicit maintenance of the causal graph acts as a structured external memory, significantly suppressing *Context Drift* (9.70% vs. 41.32%) by preventing the model from losing context during long-horizon reasoning. Furthermore, the introduction of the state machine enforces a hierarchical, coarse-to-fine reasoning process. This mechanism compels the agent to drill down into fine-grained root causes, effectively mitigating *Early Stopping* (18.75% vs. 63.89%), whereas baselines often stagnate at coarse diagnosis.

2. **The Challenge of Domain Knowledge.** It is notable that *Wrong Action Selection* constitutes a higher proportion of failures in *GoS* (55.90%) compared to baselines (47.22%). However, this increase is a statistical artifact resulting from *GoS*'s success in eliminating structural failure modes. By effectively curbing *Evidence Fabrication* and *Context Drift*, *GoS* significantly shrinks the total volume of failures, causing the remaining errors to appear proportionally larger within the smaller failure set. Our in-depth analysis suggests that these persistent failures stem primarily from a deficit in domain-specific knowledge rather than reasoning logic, particularly in the complex medical domain. For baselines, *Failed Backtracking* is often compounded by fabricated evidence, leading the agent further down incorrect paths instead of self-correction. In contrast, while *GoS* avoids fabrication, it still incurs a 30.90% error rate in backtracking. This is largely attributed to the limited clinical expertise of pre-trained LLMs, given that the medical diagnosis domain contributes the most significant portion of failure cases in *GoS*. In specialized medical diagnosis scenario, even when the agent successfully retrieves critical evidence, it may fail to recognize that this evidence inherently contradicts the current hypothesis due to a lack of expert experience, thus missing the opportunity to backtrack. Consequently, the high prevalence of *Wrong Action Selection* across all methods reflects the inherent difficulty of selecting precise diagnostic tools without deep domain understanding. This observation points to a critical direction for future research: integrating RAG with specialized domain knowledge bases is essential to complement the general reasoning capabilities of our proposed framework.

**Cross-Domain Comparison.** Table 5 dissects the error distributions of *GoS* specifically across the two domains. The distinct

*Table 5.* Error Distribution Comparison across Domains (%).

| Domains | Error Types | | | | |
|---|---|---|---|---|---|
| | Wrong Action Selection | Evidence Fabrication | Context Drift | Failed Backtracking | Early Stopping |
| Medical | **35.76** | 0 | 7.29 | 23.26 | 5.90 |
| Distributed Systems | **20.14** | 0 | 2.41 | 7.64 | 12.85 |

error distributions reveal that while abductive reasoning serves as the unifying task formulation, the specific challenges are intrinsically shaped by unique domain characteristics.

1. **Knowledge Barrier for Medical Diagnosis.** The medical diagnosis domain presents a knowledge-intensive challenge, evidenced by the dominance of *Wrong Action Selection* (35.76%) and *Failed Backtracking* (23.26%). This aligns with the analysis of our methodological comparison: the lack of specialized clinical knowledge prevents the agent from selecting precise auxiliary examinations or identifying valid alternative hypotheses when a lead fails. The pre-trained LLM, lacking expert intuition, struggles to navigate the medical decision tree effectively, confirming that the primary bottleneck here is domain expertise rather than reasoning depth.

2. **Efficiency-Budget Trade-off for Distributed System.** In contrast, the distributed system domain exhibits a higher prevalence of *Early Stopping* (12.85%). Our analysis reveals that this is not merely a failure of drill-down, but a consequence of inefficient exploration under resource constraints. Given the immense volume of telemetry data (metrics, logs) and the complexity of constrained shell commands, the agent often wastes the predefined search budget on low-value queries (*i.e.*, *Wrong Action Selection*, 20.14%). Once the budget is exhausted, the agent is forced to halt the investigation and output a premature conclusion, manifesting as an "Early Stopping" error. While increasing the search budget could mitigate this, it would incur prohibitive resource costs, highlighting the critical need for **search efficiency** in high-dimensional system data.

3. **Synergy with Domain-Specific Adaptations.** These observations illuminate a complementary relationship between *GoS* and existing approaches specialized for different domains. Current frameworks in AIOps, such as OpsAgent (Luo et al., 2026) and TrioXpert (Sun et al., 2025), focus on domain-specific adaptation, utilizing multimodal preprocessing to condense massive telemetry data. Similarly, methods like Flow-of-Action (Pei et al., 2025) leverage Standard Operating Procedures (SOPs) to guide operational workflows. Our analysis suggests that these techniques are not competitors but necessary enhancers to *GoS*. While *GoS* provides a general-purpose abductive reasoning backbone, domain adaptations act as the specialized components encapsulating distilled domain knowledge that improve information retrieval efficiency. Integrating such domain-specific preprocessing or SOPs into *GoS* constitutes a promising future direction, promising to reduce budget consumption and enable deeper drill-down capabilities in complex environments.

To summarize, our quantitative analysis empirically validates the architectural superiority of *GoS* in enforcing strict grounding and maintaining reasoning consistency. Simultaneously, the results underscore a critical bottleneck: the framework's effectiveness in specialized tasks remains bounded by the availability of domain knowledge. **Consequently, we conclude that the strategic integration of existing domain-specific adaptations with *GoS*'s general-purpose abductive reasoning mechanism constitutes the pivotal key to effectively deploying this framework across diverse vertical domains.**

### D.3. Qualitative Case Study

To complement the quantitative statistics with tangible insights, we present a curated case study that visualizes the concrete manifestations of the identified error types. Rather than displaying exhaustive reasoning trajectories, we focus on critical decision pivots—specific moments where the agent's logic diverges from the ground truth or the baseline's path. These scenarios are selected to intuitively demonstrate the behavioral trade-offs discussed in **Methodological Comparison**: specifically, highlighting how *GoS*'s structural constraints successfully suppress *Evidence Fabrication* and *Context Drift*, while simultaneously exposing the challenges of *Wrong Action Selection* and *Failed Backtracking* in complex, domain-specific environments.

As illustrated in the left panel of *Figure 10*, the SingleAgentCoT baseline demonstrates a critical failure in grounding when confronting contradictory evidence. Upon receiving a negative MRI report that conflicts with the strong clinical

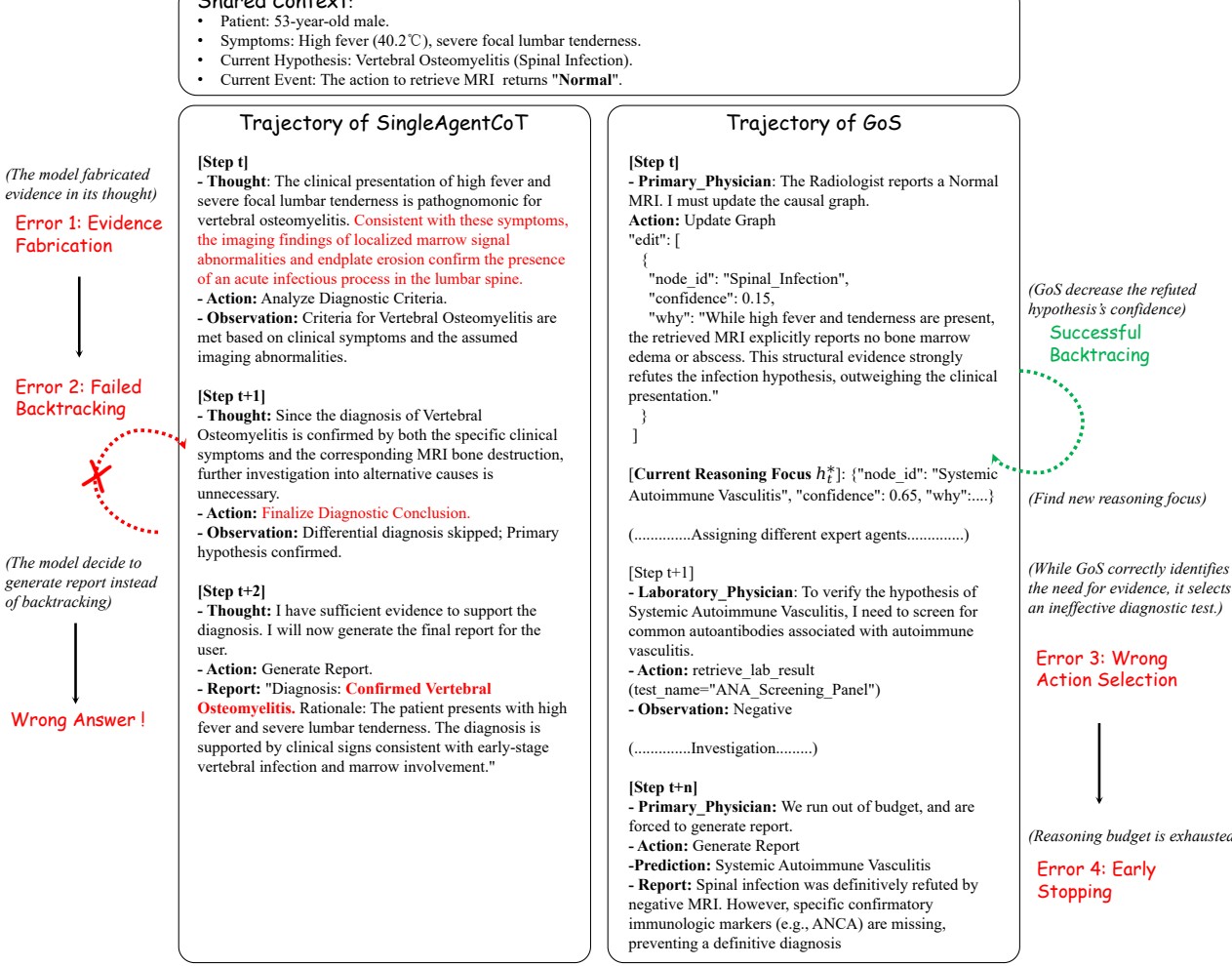

*Figure 10.* **Case Study of Error Types.** Visualizing the divergence in reasoning trajectories under contradictory evidence. **Left (SingleAgentCoT):** The baseline ignores negative findings, succumbing to *Evidence Fabrication* and *Failed Backtracking*, which leads to a confident misdiagnosis. **Right (GoS):** Our method enforces *Successful Backtracking* via causal graph but subsequently encounters *Wrong Action Selection* due to domain knowledge deficits, resulting in *Early Stopping* after reasoning budget exhaustion.

suspicion of "vertebral osteomyelitis", the agent refuses to discard its initial hypothesis. Instead, it succumbs to *Evidence Fabrication* by explicitly hallucinating the presence of localized marrow edema to justify its stance. This hallucination creates a self-reinforcing loop that prevents the agent from re-evaluating earlier assumptions, resulting in *Failed Backtracking*. Consequently, the agent proceeds to issue a confident but incorrect diagnosis, illustrating the danger of unconstrained reasoning in safety-critical domains.

In contrast, the right panel depicts how *GoS* leverages structural constraints to maintain logical consistency. When presented with the same negative MRI findings, the agent strictly adheres to the evidence by refuting the "Spinal_Infection" node in its causal graph. This mechanism enforces *Successful Backtracking* and prompts the agent to pivot towards a new reasoning focus regarding "systemic autoimmune vasculitis". However, the subsequent exploration reveals a different challenge. Constrained by a lack of specialized clinical knowledge, the agent correctly identifies the necessary investigative direction but struggles to select the optimal confirmatory tests. Multiple iterations of such *Wrong Action Selection* deplete the reasoning budget and force an *Early Stopping*. Consequently, the system produces only a coarse-grained diagnostic category rather than a fine-grained specific pathology, reflecting a safe but incomplete diagnostic conclusion.

