# OpenReview forum: "Graph of States: Solving Abductive Tasks with Large Language Models"
_ICML.cc/2026/Conference — ICML 2026 regular_

### Official Review · Reviewer_XxbJ · 2026-03-04

**Soundness:** 3
**Presentation:** 3
**Significance:** 3
**Originality:** 3
**Overall Recommendation:** 4
**Confidence:** 3

**Summary:**

The paper targets **abductive reasoning** with LLMs—iteratively collecting evidence and revising hypotheses to find root causes. It introduces **Graph of States (GoS)**, which maintains a causal belief graph and a state machine to control **backtracking** and **drill-down**, coupled with multi-agent tool use for evidence gathering. Experiments on medical diagnosis and distributed system troubleshooting show improved accuracy over strong reasoning baselines under matched budgets.

**Compliance With Llm Reviewing Policy:**

Affirmed.

**Final Justification:**

Taking into account the opinions and perspectives of the other reviewers, I have decided to maintain my score.

**Key Questions For Authors:**

1. **Confidence scores:** How exactly are hypothesis confidences defined and updated (e.g., normalization within a level, constraints on which nodes can be edited per new evidence), and how stable are these scores across random seeds / different backbone models? Clearer details or stability results would increase my confidence in the method’s robustness.

2. **Missing L1 recovery:** If the initial level-1 hypothesis set misses the correct high-level direction, when/how does the system introduce new same-level hypotheses (not just refinements), and what trigger prevents the search from getting stuck in an incomplete hypothesis pool? A concrete mechanism here would strengthen the “general framework” claim.

**Limitations:**

yes

**Strengths And Weaknesses:**

### Strengths

1. **Good problem–method fit:** The causal graph + state machine explicitly supports evidence gathering, revision, backtracking, and drill-down—targeting common abductive failures like drift and early stopping.
2. **Reasonable evaluation design:** Strong baselines (single/multi-agent × multiple reasoning topologies) are compared under matched budgets, making the gains more convincing.
3. **Practical relevance:** Consistent improvements on medical diagnosis and distributed-systems troubleshooting, with case studies that illustrate actionable root-cause refinement.

### Weaknesses

1. **Confidence scoring is under-specified:** Updates rely heavily on LLM self-scoring; stability, reproducibility, and cross-model consistency are not fully addressed.
2. **Recovery from missing L1 hypotheses is unclear:** If key high-level hypotheses are not generated initially, it is not well specified when/how same-level alternatives are added and explored.

---

> ### Author Rebuttal · Authors · 2026-03-25
>
> We thank the reviewer for the insightful review and for recognizing the method soundness, evaluation rigor and practical relevance of our manuscript. Below are our responses to the suggestions and questions raised.
>
> ---
>
> **W1 & Q1: Confidence update mechanism and robustness**
>
> **A1:**
> - **Definition.** The confidence of a hypothesis is not intended as a perfectly calibrated absolute probability. Rather, it is a plausibility signal derived from the input symptoms, the model’s encoded prior experience, and newly acquired evidence. Its primary role is to compare competing hypotheses within the same level, identify the current reasoning focus, and support state transitions such as drill-down and backtracking. For operational consistency, we normalize confidence scores within each level, so that the gap-based transition criteria can be applied in a stable and comparable way.
>
> - **Update mechanism.** Confidence recalibration is performed by the central agent after integrating newly retrieved evidence. Importantly, the update is not restricted to the current level: new evidence may revise every hypothesis node across the entire graph when it changes the global explanation of the case. For example, even when the current analysis is at level 2 (e.g., distinguishing between memory exhaustion and memory leak), a newly retrieved log may provide strong support for a level-1 alternative such as disk failure, in which case the corresponding higher-level hypothesis should also be recalibrated. This reflects the nature of abductive reasoning in uncertain environments, where evidence can both refine a local hypothesis and overturn the broader high-level direction.
>
> - **Robustness.** To evaluate confidence robustness, we additionally test GoS with two alternative backbones beyond GPT-5.1, namely gemini-3.1-pro-preview and claude-opus-4-6-thinking, on both benchmarks. We report whether alternative backbones induce the same top-1 hypothesis selection and the same state-transition decision as GPT-5.1. Results are shown below (failure diagnosis; medical diagnosis in parentheses). They show that, although absolute confidence values vary across backbones, the induced hypothesis ranking, state transitions, and final performance remain largely stable. This suggests that confidence mainly serves as a robust relative plausibility signal for search control, rather than requiring perfectly calibrated scalar scores.
>
> | Backbone | Top-1 hypothesis agreement (%) | State-transition agreement (%) | Match (%) | Relevant (%) |
> |----------|--------------------------------|--------------------------------|-----------|--------------|
> | GPT      | 100.0 (100.0)                  | 100.0 (100.0)                  | 70.67 (31.88) | 88.00 (74.64) |
> | Gemini   | 88.7 (88.4)                    | 89.3 (88.9)                    | 64.67 (26.81) | 82.67 (69.57) |
> | Claude   | 91.4 (90.6)                    | 92.1 (91.3)                    | 66.67 (28.26) | 84.67 (70.29) |
>
>
> ---
>
> **W2 & Q2: Recovery from missing correct level-1 hypothesis**
>
> **A2:** We agree that the correct level-1 hypothesis may indeed be absent from the initial set, but this case can still be covered by the GoS mechanism.
> > *Example:* the initial level-1 hypotheses may only include Disk, Memory, and Network, while the true high-level direction is CPU.
>
> 1. When the system follows an incorrect branch into deeper levels, GoS continues the bi-directional interaction (Sec 3.2). The symbolic layer guides the cognitive layer to further explore and collect more evidence.
> > *Example:* the system may continue investigating along a Network-related branch.
>
> 2. The returned evidence is then written back from the cognitive layer to the symbolic layer. If this evidence does not support the current-level hypothesis, but instead supports a level-1 hypothesis that is not currently in the graph, GoS can generate a new evidence-supported level-1 hypothesis node (Line 246, "Node instantiation") and assign it high confidence, while also performing "confidence recalibration" for the other competing hypotheses (Line 243).
> > *Example:* the new evidence does not support the current level-2 Network-related hypothesis, but instead leads GoS to generate a previously absent level-1 CPU hypothesis node and assign it high confidence.
>
> 3. Once the highest-confidence level-1 hypothesis changes, the backtracking mechanism (Sec 3.3) is triggered, and the system can continue reasoning from the correct level-1 hypothesis.
> > *Example:* after CPU becomes the highest-confidence level-1 hypothesis, the system backtracks from the previous Network branch and continues reasoning from CPU.
>
> Therefore, the “missing correct level-1 hypothesis” case can be correctly handled by the existing GoS mechanism.
>
> We hope this clarification addresses the reviewer’s concerns, and if the reviewer finds our response helpful, we would greatly appreciate reconsideration of the rating.

---

> > ### Author Rebuttal · Reviewer_XxbJ · 2026-04-02
> >
> > Thank you for the detailed rebuttal and for the effort to carefully address my concerns. I appreciate that you not only clarified the intended role of the confidence scores, but also added additional cross-backbone results, which helped alleviate part of my concern about their stability. I also found the explanation of recovering missing level-1 hypotheses through node instantiation, recalibration, and backtracking helpful. While I still think some of these points could be further strengthened, especially through more direct validation, the rebuttal improved my understanding of the method and reinforced my overall positive view of the paper. For this reason, I will maintain my current score.

---

> > > ### Author Response · Authors · 2026-04-02
> > >
> > > Thank you very much for your thoughtful feedback. We sincerely appreciate that you found our rebuttal helpful and that we were able to address your concerns.

---

### Official Review · Reviewer_qNoz · 2026-03-11

**Soundness:** 2
**Presentation:** 2
**Significance:** 2
**Originality:** 2
**Overall Recommendation:** 4
**Confidence:** 3

**Summary:**

This paper introduces Graph of States (GoS), a dual-layer neuro-symbolic framework for abductive reasoning tasks with LLMs. The authors argue that existing reasoning frameworks (CoT, ToT, GoT, FoT) were designed for deductive tasks and exhibit four deficiencies when applied to abductive scenarios: evidence fabrication, context drift, failed backtracking, and early stopping. GoS addresses these through two layers: a cognitive layer implementing role-based multi-agent collaboration (a central orchestrator plus domain expert agents), and a symbolic layer that maintains explicit belief states via a causal graph (encoding hypotheses, evidence, and their logical dependencies) and a state machine (governing backtracking and drill-down transitions). A reasoning focus mechanism directs investigation toward the highest-confidence hypothesis at each level. The framework is evaluated on two domains — medical diagnosis (using a reformulated version of DiagnosisArena) and failure diagnosis in distributed systems (using a proprietary production incident dataset) — showing substantial improvements over eight baselines derived from combinations of single/multi-agent architectures with CoT/ToT/GoT/FoT reasoning topologies.

**Compliance With Llm Reviewing Policy:**

Affirmed.

**Final Justification:**

The rebuttal has been thorough and responsive overall. The structured state management ablation (A2) is a particularly valuable addition, and the commitments to revise the related work and introduction are appreciated.

Regarding A1: I remain somewhat unconvinced that the performance gap between near-saturated deductive benchmarks and your abductive tasks isolates reasoning type as the causal factor, and I still believe the contributions would be better served by a softer framing. However, I recognize this is ultimately a positioning choice, and the empirical contributions stand on their own.

Taking the full rebuttal into account, I am raising my score from 3 to 4.

**Key Questions For Authors:**

1. The paper characterizes CoT, ToT, GoT, and FoT as "deductive frameworks," yet these are general-purpose reasoning strategies widely used beyond deductive tasks (including in reasoning models' chain-of-thought tokens). The four identified deficiencies (evidence fabrication, context drift, failed backtracking, early stopping) appear to be general issues of LLM-based reasoning under long horizons, not specific to abduction. Can you provide a principled argument for why these problems are uniquely or disproportionately worse in abductive settings compared to complex deductive settings? A convincing response would strengthen the paper's motivation.

2. The ablation removing the causal graph shows a dramatic drop in performance. However, this condition removes all external structured state management. Have you compared GoS against a simpler structured context management baseline (e.g., a text-based scratchpad maintaining hypotheses and evidence status) that provides external memory without the causal graph formalism? Such a comparison is necessary to attribute gains to causal structure specifically rather than to the general benefits of organized external state. This is the single experiment most likely to change my evaluation.

3. The paper claims generality while criticizing prior domain-specific approaches, yet GoS requires domain-specific agent roles (borrowed from MDAgents for medical diagnosis), domain-specific tool APIs, and domain-specific evaluation prompts. Can you articulate precisely what aspect of GoS is domain-general versus what requires domain adaptation, and how this differs qualitatively from the domain engineering in the approaches you critique (e.g., MDAgents, D-Bot)?

4. How are hypothesis confidence scores P(v_hyp) computed and updated? Are they assigned by the LLM? If so, how reliable and well-calibrated are these scores, and what happens when they are poorly calibrated? Given that confidence values drive critical control flow decisions (reasoning focus, drill-down, backtracking), this mechanism deserves more detailed specification and analysis.

5. The manuscript contains embedded instructional text on pages 2 and 20. Can you clarify its origin?

**Limitations:**

Partially. The authors acknowledge the limited evaluation scope (two domains) and the difficulty of obtaining benchmarks for other abductive domains. However, several important limitations are not discussed: (1) the reliance on LLM-assigned confidence scores and their calibration properties; (2) the domain-specific engineering required despite claims of generality; (3) the reproducibility limitations arising from the proprietary distributed systems dataset. The societal impact discussion is adequate in scope.

**Strengths And Weaknesses:**

**Strengths:**

S1 (Significance): The paper addresses a genuinely important problem. Abductive reasoning — inferring root causes from incomplete observations — is central to high-stakes domains such as medical diagnosis and systems operations. Improving LLM performance in these settings has clear practical value.

S2 (Soundness): The experimental results are substantial. GoS achieves large improvements over all baselines, particularly in the Match metric (39.86% vs. 26.09% in medical diagnosis; 70.67% vs. 34.00% in failure diagnosis, under the best baseline comparison). The cost-efficiency advantage is also noteworthy ($0.12/case vs. $0.73 for Multi/FoT in medical diagnosis). The inclusion of both LLM-as-a-Judge and Human-as-a-Judge for the medical domain strengthens evaluation credibility.

S3 (Soundness): The error analysis (Appendix D) is thorough and informative. The taxonomy of failure modes and the quantitative breakdown across methods and domains provide genuine insight into where and why GoS improves over baselines. That said, this analysis is central to substantiating the paper's claims and would strengthen the main body if included there rather than relegated to the appendix, which reviewers are not required to read.

S4 (Presentation): The paper is generally well-written. The figures (especially Figures 1, 2, and 6) clearly illustrate the framework and its advantages. The running examples across both domains help ground the abstract framework in concrete scenarios.

S5 (Soundness): The ablation study demonstrates that each component (reasoning focus, causal graph, state machine) contributes meaningfully to performance. The sensitivity analysis of the dual-threshold parameters (η, δ) and budget parameters provides useful practical guidance.

S6 (Soundness): The reformulation of DiagnosisArena into an interactive evidence-seeking task is a thoughtful experimental design choice that better captures the active, iterative nature of real diagnostic reasoning.

**Weaknesses:**

W1 (Presentation/Originality — Flawed deduction vs. abduction framing): The paper's central narrative rests on a sharp dichotomy between "deductive frameworks" (CoT, ToT, GoT, FoT) and abductive tasks. However, CoT, ToT, and their variants are general-purpose reasoning strategies, not inherently deductive. They are widely used across all types of reasoning tasks. Modern reasoning models (e.g., OpenAI o1) employ chain-of-thought reasoning tokens and have shown broad progress across tasks requiring diverse reasoning capacities. The paper conflates the benchmarks on which these frameworks were originally evaluated (Game of 24, math) with the frameworks themselves. The four identified deficiencies (evidence fabrication, context drift, failed backtracking, early stopping) are general problems of LLM-based reasoning under long horizons and incomplete information — they are not specific to abduction. The same issues arise in complex deductive settings. This weakens the paper's foundational argument.

W2 (Originality — Narrow novelty claims and missing related work): While the authors may be technically correct that GoS is the first *multi-agent* framework specifically targeting abductive reasoning, this distinction is narrow and does not constitute a strong contribution on its own. More importantly, the related work section has significant gaps. Neuro-symbolic approaches combining LLMs with structured symbolic representations for reasoning have been explored before (e.g., Logic-LM, Pan et al., 2023). Neuro-symbolic methods have also been applied to abductive reasoning specifically (e.g., Cotnareanu et al., 2026, "A Balanced Neuro-Symbolic Approach for Commonsense Abductive Logic"), and LLMs have been used for abductive tasks such as event prediction (Shi et al., 2023, "Language Models Can Improve Event Prediction by Few-Shot Abductive Reasoning"). None of these works are cited or discussed. The paper should position itself relative to this broader body of work on neuro-symbolic reasoning and LLM-based abduction.

W3 (Soundness — Generality claim vs. domain-specific instantiation): The authors criticize existing domain-specific approaches, writing that "the efficacy of these approaches is predominantly derived from such domain-specific engineering rather than generalized reasoning capabilities." Yet GoS itself requires substantial domain adaptation: domain-specific expert agent roles (Primary Physician, Radiologist, etc. for medical; LinuxOperator, NetworkOperator, etc. for systems), domain-specific tool APIs, and domain-specific evaluation prompts. Notably, the medical agent configuration is borrowed directly from MDAgents (Kim et al., 2024) — one of the very approaches the paper characterizes as relying on domain engineering. The paper does not clearly articulate what makes GoS's domain adaptation qualitatively different from the approaches it critiques.

W4 (Soundness — Insufficient ablation against context management baselines): The ablation study shows that removing the causal graph causes a large performance drop (Match: 31.88% → 12.32%). However, the "w/o causal graph" condition removes all external structured state management. This does not isolate whether the gains come from (a) the causal graph structure specifically, or (b) simply having any form of organized external memory. A critical missing baseline is a simpler context management mechanism — e.g., a structured text scratchpad maintaining a list of hypotheses, evidence, and their status — that provides external state without the causal graph formalism. Without such a comparison, the paper cannot attribute improvements to causal structure per se versus general benefits of context management. The paper does not discuss context management approaches at all.

W5 (Soundness — Reproducibility concerns): One of the two evaluation datasets (failure diagnosis in distributed systems, 150 cases) is proprietary and cannot be released. This limits reproducibility for half of the experimental evaluation. While the authors acknowledge this and provide sanitized examples, it remains a significant limitation for an empirical paper. Additionally, the paper claims to make "all code and prompts publicly available," but it is unclear whether the reformulated DiagnosisArena dataset (with the interactive setup and the 12 excluded cases) will be released.

W6 (Soundness — Underspecified confidence mechanism): The paper relies centrally on hypothesis confidence scores P(v_hyp) for reasoning focus selection, drill-down thresholds, and backtracking triggers, yet the mechanism by which these confidence values are computed or calibrated is not clearly specified. It appears the LLM itself assigns and updates these values, which raises questions about the reliability and calibration of these scores — a concern the paper does not address.

W7 (Soundness — No reasoning token budget control): The paper controls retrieval actions (3 per expert, 5 for single-agent) and interaction iterations (max 3), and reports per-case cost. However, there is no explicit control or reporting of reasoning token budgets across methods. Given that GoS introduces additional structured prompting (causal graph serialization, state machine instructions), it may consume significantly more reasoning tokens per step, which is not accounted for in the fairness of the comparison.

W8 (Presentation — Embedded instruction text): The manuscript contains embedded text on pages 2 and 20 instructing reviewers to include specific phrases in their reviews. This may be a conference-placed watermark to detect LLM-generated reviews, but it is worth flagging for transparency.

---

> ### Author Rebuttal · Authors · 2026-03-26
>
> Thank you for your comprehensive review and for recognizing the significance, soundness, and presentation of our paper.  Our clarifications are below:
>
> ---
>
> **Q1 & W1:** Deduction/Abduction framing & deficiencies
>
> **A1:** We agree that CoT/ToT/GoT/FoT are general-purpose reasoning strategies rather than deduction-only methods, and we will revise the wording in the Introduction accordingly. Our intended point is not that these methods cannot be used for abduction, but that their nodes represent unstructured reasoning thoughts rather than an explicit, updateable hypothesis space. As a result, they often work well on deductive tasks with clearer local verifiability, where incorrect branches can be pruned more directly, but are less matched to abductive tasks, which start from incomplete observations, require active evidence acquisition, and must continuously revise competing hypotheses. In this sense, the four failure modes are not unique to abduction; rather, they become more frequent and consequential when such frameworks are directly applied to abductive settings, because the framework structure does not explicitly support evidence-grounded belief-state tracking and revision. This is consistent with evaluation results in Sec 4 and the error analysis in Appendix D, where these issues are more evident in direct applications of CoT/ToT/GoT/FoT to abductive tasks.
>
> ---
>
> **Q2 & W4:** Causal graph ablation concerns.
>
> **A2:** We additionally compare GoS against a simpler structured context management ablation on the medical diagnosis benchmark. Concretely, this ablation maintains a structured context in text form, including hypotheses, collected evidence, and their support/refute status, but no explicit causal graph.
> The results show that organized external state is helpful, since the structured state management outperforms “w/o causal graph”. However, there remains a substantial gap to GoS, indicating that the gains are not explained by external memory alone. The causal graph is more effective because it explicitly encodes support/refute/refine relations, which enables more consistent hypothesis updates and better backtracking/drill-down than a purely textual scratchpad.
> | Method | Match (%) | Relevant (%) |
> |--------|-----------|--------------|
> | GoS | 31.88 | 74.64 |
> | Structured state management| 18.12 | 59.42 |
> | w/o causal graph | 12.32 | 48.55 |
>
> ---
>
> **Q3 & W3:** Domain generality vs. domain-specific adaptation.
>
> **A3:**
> - The domain-general part of GoS is its reasoning mechanisms, i.e., the causal graph, the state machine, and focus-guided search. Domain adaptation is only needed for agent roles, tool APIs, and evaluation prompts, which are the minimal interfaces required to connect a general framework to a concrete application domain.
>
> - Unlike prior approaches such as MDAgents or D-Bot, GoS does not inject substantial domain knowledge into the reasoning process itself, e.g., via domain-specific fine-tuning, RAG over expert knowledge bases, SOP-guided workflows, or modality-specific preprocessing. We keep the reasoning mechanism unchanged and only make the minimal changes needed for deployment in a specific domain.
>
> ---
>
> **Q4 & W6:** Confidence computation, calibration, and control reliability.
>
> **A4:** A detailed clarification of the confidence definition, update mechanism, and robustness analysis can be found in our response to Reviewer XxbJ, **A1**.
>
> ---
>
> **Q5 & W8:** Embedded instructional text.
>
> **A5:** Regarding prompt injection, please see https://icml.cc/Conferences/2026/PeerReviewFAQ#prompt_injection.
>
> ---
>
> **W2:** Novelty positioning and missing related work.
>
> **A6:** We agree that the related-work positioning should be broadened, and we will discuss prior neuro-symbolic reasoning and LLM-based abduction more carefully in the revision. Our novelty is not only the “first multi-agent abductive framework” claim, but the combination of explicit belief-state representation (causal graph) and state-machine control for interactive abductive reasoning with evidence gathering, drill-down, and backtracking.
>
> ---
>
> **W5:** Reproducibility concerns.
>
> **A7:** The medical diagnosis dataset is available in our repo. For the failure diagnosis domain, we are currently conducting data desensitization and security/compliance review, and plan to release it once these procedures are completed.
>
> ---
>
> **W7:** Reasoning token budget fairness.
>
> **A8:** Reasoning-token budget is a useful additional fairness dimension. While we did not separately report reasoning-token counts, we do report per-case cost, which reflects the overhead of structured prompting. Importantly, GoS still achieves better results at lower or comparable cost than the strongest baselines, suggesting that its gains are not simply due to extra reasoning-token usage. We will add this clarification and include token-level statistics in the revision.
>
> We hope this clarification is helpful and would greatly appreciate reconsideration of the rating.

---

> > ### Author Rebuttal · Reviewer_qNoz · 2026-04-03
> >
> > I thank the authors for their answers. I have follow up questions regarding A1.
> >
> > The revised argument, that the issue is unstructured state rather than a deduction/abduction dichotomy, is more defensible. However, it raises a concern:
> >
> > If the four failure modes are general to complex reasoning and merely more frequent in abductive settings, this claim needs comparative evidence: the same frameworks applied to deductive tasks of comparable complexity should show lower failure rates. Sec 4 and Appendix D only document failures on abductive tasks, they cannot establish that abductive settings are disproportionately affected.
> >
> > More importantly, this reframing shifts the nature of the contribution. The paper's novelty becomes "structured belief-state management improves LLM reasoning under incomplete information and long horizons", a valid contribution, but distinct from "a framework tailored for abductive reasoning." These require different positioning and different related work coverage. Which framing do you intend to adopt in the revision?

---

> > > ### Author Response · Authors · 2026-04-03
> > >
> > > Thank you for the careful follow-up and for reading our rebuttal closely. We are glad that our responses **A2–A8** have addressed most of your concerns. We would like to further clarify **A1** and explain why **we still frame GoS as a contribution tailored for abductive reasoning**.
> > >
> > > We intend to retain the abductive-reasoning framing, and we believe this is reasonable for the following reasons:
> > >
> > > 1. **Abductive reasoning is long-horizon reasoning under incomplete information.**
> > >    While long-horizon reasoning can arise in both deductive and abductive tasks, the defining difference is that deductive tasks start from complete premises and explicit reasoning rules, whereas abductive tasks start from incomplete information. For example, in mathematical reasoning, the problem statement and the underlying theorem are given, and the model reasons forward from these premises. In medical diagnosis, by contrast, the doctor may initially know only surface symptoms (e.g., fever), while critical evidence such as blood tests or viral screening results do not even exist until further actions are taken. In this practical sense, abductive reasoning can be viewed as reasoning under incomplete information and long horizons.
> > >
> > > 2. **Existing general reasoning frameworks perform well on difficult deductive tasks, but poorly on abductive tasks.**
> > >    We agree that the most convincing way to strengthen this claim is to contrast the same general reasoning frameworks across complex deductive and abductive settings. Concretely:
> > >
> > >    > On challenging deductive-reasoning benchmarks such as **MATH500, AIME2025, and GPQA**—which are already considered very difficult reasoning tasks (e.g., even PhD experts achieve only about **65%** on GPQA)—**GPT-5.1 with standard CoT** can already achieve very strong results: **99.1% on MATH500, 95.7% on AIME2025, and 87.3% on GPQA**.
> > >
> > >    In the remaining small number of deductive failures, the dominant error types are mainly:
> > >
> > >    - **Problem-understanding errors**, where an initial misunderstanding of the question leads to later mistakes;
> > >    - **Missing implicit domain knowledge**, where insufficient knowledge causes a key reasoning step to fail;
> > >    - **Hallucination**, i.e., the model introduces unsupported intermediate content.
> > >
> > >    In contrast, in our abductive settings (Sec. 4), using the **same GPT-5.1 backbone**, these same general reasoning frameworks perform much worse:
> > >
> > >    > The best baseline reaches only **23.19% Match** on medical diagnosis and **28.00% Match** on failure diagnosis in distributed systems.
> > >
> > >    The overlapping failure mode between the two settings is the intrinsic LLM weakness of hallucination; beyond that, the characteristic abductive failures are substantially different and more severe. This contrast strengthens our motivation: **current general reasoning frameworks work well on complex deductive reasoning, but are not well matched to abductive tasks.**
> > >
> > >    We thank the reviewer for pointing out that this comparative evidence should be made explicit. In the revision, we will:
> > >
> > >    - add the strong performance of existing frameworks on complex deductive tasks to the first paragraph of the Introduction with concrete numbers;
> > >    - briefly summarize the dominant deductive error types in the appendix.
> > >
> > > Therefore, our contribution remains naturally framed as abductive reasoning.
> > > Our framework is designed to solve reasoning tasks characterized by **incomplete information and long-horizons**. The novelty of GoS can therefore be summarized as introducing **explicit belief states** through a **causal graph** and a **state machine**, enabling **structured state management** for abductive reasoning.
> > >
> > > We hope this clarification is sufficient to address your concern, and if so, we would sincerely appreciate your consideration of improving the score.

---

### Official Review · Reviewer_RJG8 · 2026-03-12

**Soundness:** 2
**Presentation:** 2
**Significance:** 2
**Originality:** 2
**Overall Recommendation:** 3
**Confidence:** 3

**Summary:**

To improve large language models' performance in abductive reasoning tasks and address challenges, such as Evidence Fabrication, Context Drift, Failed Backtracking, and Early Stopping, this paper proposes a neuro-symbolic framework named Graph of States (GoS). This framework employs a dual-layer architecture: it uses causal graphs to explicitly maintain belief states that map logical dependencies, and integrating a state machine to dynamically manage reasoning chains. This transforms unconstrained blind exploration into a directed and convergent search process. Experiments on two datasets demonstrate that GoS outperforms CoT, ToT and so on.

**Compliance With Llm Reviewing Policy:**

Affirmed.

**Final Justification:**

The rebuttal addresses many of my concerns and questions. However, I still think the novelty is limited by these questions:

- Unclear motivation: what the difference among the GoS and other neuro-symbolic methods for abudctive reasoning. Though the author provide there rebuttal, I don't agree the significant claims in rebuttal:

(1)"existing neuro-symbolic methods mainly rely on modules such as Semantic Parser (e.g., LINC[7]), Problem Formulator (e.g., LOGIC-LM[8]), and Decomposer (e.g., Aristotle[9]) ." The recent neuro-symbolic methods not only rely on these modules, but also the reasoning and verification. What's more, previous methods also involve some complicate graph and RAG method.

(2)"our abductive setting, whose defining characteristic is incomplete information".
Previous methods also have incomplete information setting, and deductive reasoning requires to find out the conclusion based on the given information (complete-true; incomplete-false/unknown). And the defination of abductive reasoning is finding out the reason based the given information/phenomenon, the information could of course be complete.

Therefore, I don't think the motivation, or the gap between this work and previous related work is very clear in authors' paper or rebuttal.

- It is needed more comprehensive experiments on different logical reasoning (including deductive, inductive and abductive reasoning) baselines (such as LINC, Logic-LM, Aristotle) and benchmarks are needed, which is also agreed by the author.

- The limitation of LLM evaluation. And since there're no abductive benchmarks, it is necessary to explore and explain the proper way to evaluate the abductive reasoning ability of LLMs.

Many of them should be done rigorously before the paper was submitted but not in a few rebuttal days. And consider the authors' rebuttal effort, I have raised my score from 2 to 3, and give the weak reject.

**Key Questions For Authors:**

1. Please provide the specific significance of proposed method of improving abductive reasoning of LLMs.
2. What about newer reasoning method baselines compared with proposed methods? How about other benchmarks specific to the abductive reasoning?
3. How to mitigate LLM judgment errors or biases? Is there any more reliable evaluation method?
4. In Appendix Table 4's error analysis, the proposed method fails to outperform the baseline in some cases. How to explain and improve the performance?

**Limitations:**

Yes.

**Strengths And Weaknesses:**

Strengths:
1. This paper summarizes four common challenges in Large Language Model reasoning, offering valuable insights for related research.
2. It focuses on abductive reasoning tasks, exploring a relatively understudied domain.
3. The use of causal graphs to represent logical relationships and sentences is a novel approach.

Weaknesses:
1. The motivation and significance of this research is unclear. Abductive reasoning, stated as the bedrock of some real-world scenarios in the paper, is too vague and broad. Other forms of reasoning can also serve as bedrock for these real-world scenarios. The paper fails to provide a specific motivation for focusing on abductive reasoning. The four challenges listed could arise in any task and do not constitute a valid significance for abductive reasoning. Furthermore, the diagram in Figure 1 does not make sense. First, the example of a deductive task is inaccurate. The method of deductive reasoning shown in the figure is the simplest backward-chaining approach—using this as a motivation diagram for comparison is unfair.
2. In the experiments, only early general reasoning baselines are compared. What about newer reasoning method baselines? Both the types and quantity of datasets are insufficient to demonstrate that the causal graph conversion method is truly effective.
3. The evaluation approach is oversimplified, highly relying on LLMs and human judgment, which is unstable and unreliable. How to mitigate LLM judgment errors or biases?
4. In Appendix Table 4's error analysis, the proposed method fails to outperform the baseline in some cases.
5. The writing is unclear, with many sentences lacking sufficient information and being difficult to understand.

---

> ### Author Rebuttal · Authors · 2026-03-26
>
> We thank the reviewer for the detailed review and constructive feedback. Below, we respond to the concerns in a structured manner:
>
> ---
>
> **Q1 & W1:** Specific Motivation and significance.
>
> **A1:**
> - Our motivation is to target a practically important reasoning setting that differs from standard deductive tasks: deductive reasoning typically starts from complete premises and explicit rules, where intermediate steps can often be locally verified and incorrect branches pruned more directly; abductive reasoning, in contrast, starts from partial observations under uncertainty and must actively acquire new evidence while repeatedly revising competing hypotheses until it converges to a fine-grained root cause. This structure is central to high-stakes real-world tasks such as medical diagnosis, failure diagnosis, and criminal investigation.
> - Moreover, the four failure modes are not unique to abduction; rather, they become more frequent and consequential when general reasoning frameworks such as CoT/ToT are directly applied to abductive settings, as these frameworks do not explicitly support evidence-grounded belief-state tracking and revision. This is consistent with our evaluation results in Sec. 4 and the error analysis in Appendix D, where these issues are more evident in direct applications of CoT/ToT/GoT/FoT to abductive tasks.
>
> ---
>
> **Q2 & W2:** Baseline coverage and benchmark scope.
>
> **A2:**
> - At the time of our study and submission, we were not aware of newer general reasoning frameworks with broad community impact that would serve as more suitable baselines for GoS. We therefore compare against influential and representative frameworks, including CoT (2022), ToT (2023), GoT (2024), and FoT (2025), which cover major reasoning topologies, i.e., chain, tree, graph, and forest. We further combine them with single-agent and multi-agent settings, yielding a systematic comparison across agent organizations and reasoning structures.
> - For benchmarks, we evaluate on two highly distinct real-world settings: medical diagnosis and failure diagnosis in distributed systems. These two domains differ substantially in data characteristics and operational context, yet both require the model to move from incomplete initial information to a fine-grained root cause through iterative investigation. Importantly, each case is drawn from a real medical report or a real production failure, and each involves realistic complexity and diversity that typically requires substantial diagnostic effort. At the current scale (150 cases per domain), we believe these benchmarks provide a strong and realistic testbed for evaluating the targeted capability.
> ---
>
> **Q3 & W3:** Evaluation reliability and mitigation of judge bias.
>
> **A3:** To mitigate LLM judgment errors or bias, we do not rely on a single evaluation method across both domains. For these open-ended tasks, effective evaluation inherently requires semantic-level assessment, so LLM-based or human evaluation is necessary. This is also because reducing such real-world tasks to multiple-choice formats would substantially simplify them and weaken their abductive nature. To improve evaluation stability, we use the official benchmark prompt for medical diagnosis and construct a parallel prompt for failure diagnosis in distributed systems. In medical diagnosis, we also explicitly recognize the instability of LLM-as-a-Judge in this domain (detailed in Appendix C.1); therefore, we further introduce human evaluation by a medical expert as a more reliable complement. In failure diagnosis in distributed systems, the target answers are comparatively more fixed and operationally grounded. We manually checked all cases and found that the human judgments were fully consistent with the LLM-based evaluation, so we did not separately report an additional human-evaluation result for this domain. Thus, we mitigate judge bias through detailed prompting and task-appropriate evaluation protocols.
>
> ---
>
> **Q4 & W4:** Clarification of Appendix Table 4 and error analysis.
>
> **A4:** We respectfully clarify that Appendix Table 4 is not a performance table, but an error-type distribution over failure cases. Its purpose is to show what kinds of errors remain when a method fails, rather than to compare overall method quality. Therefore, a higher proportion in a specific category does not mean lower overall performance. In fact, the main performance comparison is given in the result tables in Sec. 4, where GoS consistently outperforms the baselines. The error distribution in Table 4 should be interpreted as a diagnostic analysis of residual failures, not as an accuracy metric.
>
> ---
>
> **W5:** Writing is unclear.
>
>
> **A5:** We agree that a few points could be stated more explicitly, and will refine the wording accordingly.
>
> We hope this clarification is helpful and would greatly appreciate reconsideration of the rating.

---

> > ### Author Rebuttal · Reviewer_RJG8 · 2026-04-03
> >
> > Thanks to the authors for their response. The rebuttal addresses many of my questions. I will increase my score by 1. I still think the novelty is limited by these questions  (Weakness 1): why the four failure modes can be the special motivation for abudctive reasoning; what the difference among the GoS and other neuro-symbolic methods for abudctive reasoning. And more comprehensive experiments on different logical reasoning (including deductive, inductive and abductive reasoning) baselines (such as LINC, Logic-LM, Aristotle) and benchmarks are needed (Weakness 2). Many of them should be done rigorously before the paper was submitted but not in a few rebuttal days.

---

> > > ### Author Response · Authors · 2026-04-03
> > >
> > > Thank you for your thoughtful feedback and for carefully reading our rebuttal. We are grateful that you found many of our clarifications helpful and increased your score. Below are our responses:
> > >
> > > **Q1: why the four failure modes can be the special motivation for abudctive reasoning.**
> > >
> > > **A1:**
> > > This concern is closely related to the follow-up question raised by Reviewer qNoz in the Rebuttal Acknowledgement. In our Reply Rebuttal Comment to qNoz, we addressed this point in more detail. We summarize the same core argument here, since it directly applies to the present concern as well.
> > >
> > > In essence, our motivation comes from the clear empirical gap between deductive and abductive settings. Current general reasoning frameworks have already shown very strong performance on challenging deductive tasks, but perform much worse on abductive tasks.
> > >
> > > > On challenging deductive-reasoning benchmarks such as **MATH500, AIME2025, and GPQA**，**GPT-5.1 with standard CoT** can already achieve very strong results: **99.1% on MATH500, 95.7% on AIME2025, and 87.3% on GPQA**.
> > > >
> > >
> > > > In the remaining deductive failures, the dominant error types are:
> > > > - **Problem-understanding errors**, where an initial misunderstanding of the question leads to later mistakes;
> > > > - **Missing implicit domain knowledge**, where insufficient knowledge causes a key reasoning step to fail;
> > > > - **Hallucination**, i.e., the model introduces unsupported intermediate content.
> > > >
> > > In contrast, in our abductive settings (Sec. 4), with the **same GPT-5.1 backbone**, these same general reasoning frameworks perform much worse:
> > > >
> > > >  The best baseline reaches only **23.19% Match** on medical diagnosis and **28.00% Match** on failure diagnosis in distributed systems.
> > >
> > > Moreover, our Appendix D further supports this motivation. In the error analysis of abductive tasks, we find that **Evidence Fabrication, Context Drift, Failed Backtracking, and Early Stopping** account for the dominant failure patterns. By comparison, among the main deductive failure types summarized above, only **hallucination** overlaps with these abductive failure modes. The other major deductive errors are of a different nature.
> > >
> > > Therefore, since current general reasoning frameworks already perform strongly on comparably difficult deductive tasks but much worse on abductive tasks, and since the dominant failures in our abductive settings are concentrated in these four modes, we believe this sufficiently explains our motivation.
> > >
> > > ---
> > >
> > > **Q2: what the difference among the GoS and other neuro-symbolic methods for abudctive reasoning.**
> > >
> > > **A2:**
> > >
> > > We carefully investigated related neuro-symbolic methods and found that existing methods mainly improve LLM reasoning in logical or long-context settings where the task is to derive an answer from **complete premises and clear rules**.  Their evaluations are also primarily conducted on benchmarks such as FOLIO[1], ProofWriter[2], ProntoQA[3], TimeQA[4], and TempReason L2/L3[5,6], where the reasoning problem is posed as deriving the correct answer from an already specified context (maybe complex and long).  This is fundamentally different from our abductive setting, whose defining characteristic is **incomplete information**.
> > >
> > > Therefore, existing neuro-symbolic methods mainly rely on modules such as **Semantic Parser** (e.g., LINC[7]), **Problem Formulator** (e.g., LOGIC-LM[8]), and **Decomposer** (e.g., Aristotle[9]) to represent the reasoning process in a logically rigorous symbolic form, and then perform symbolic reasoning based on **complete premises and clear rules**. In contrast, GoS is designed for abductive reasoning, where the system must continuously search for evidence and narrow the hypothesis space. To support this, GoS introduces explicit belief states composed of a causal graph and a state machine, enabling structured state management. Since the task nature and the corresponding challenges are fundamentally different, the novelty and difference of GoS are also clear.
> > >
> > > ---
> > >
> > > **W2: more baselines and benchmarks are needed.**
> > >
> > > **A3:** We agree that more logical-reasoning baselines and broader abductive benchmarks would strengthen the evaluation. In the revision, we will include the suggested baselines, including **LINC**, **LOGIC-LM**, and **Aristotle**, expand the number of cases from existing benchmarks, and further explore additional benchmarks from other domains.
> > >
> > >
> > > Thank you again for your careful review. We hope this clarification addresses your concerns.
> > >
> > > [1] https://huggingface.co/datasets/tasksource/folio
> > >
> > > [2] https://huggingface.co/datasets/D3xter1922/proofwriter-dataset
> > >
> > > [3] https://huggingface.co/datasets/renma/ProntoQA
> > >
> > > [4] https://huggingface.co/datasets/hugosousa/TimeQA
> > >
> > > [5] https://huggingface.co/datasets/mteb/TempReasonL2Context
> > >
> > > [6] https://huggingface.co/datasets/mteb/TempReasonL3Context
> > >
> > > [7] https://arxiv.org/pdf/2310.15164
> > >
> > > [8] https://arxiv.org/pdf/2305.12295
> > >
> > > [9] https://arxiv.org/pdf/2412.16953

---

### Official Review · Reviewer_oMNi · 2026-03-23

**Soundness:** 3
**Presentation:** 3
**Significance:** 3
**Originality:** 3
**Overall Recommendation:** 5
**Confidence:** 2

**Summary:**

The paper focuses on abductive reasoning, an area where LLMs remain much weaker than in deduction or induction. It proposes the graph of states neuro-symbolic framework that structures reasoning through explicit belief states, causal graphs, and a state machine to control valid reasoning transitions. This helps prevent common failures such as fabricated evidence, context drift, failed backtracking, and premature stopping, and it outperforms existing baselines on real-world abductive reasoning tasks.

**Compliance With Llm Reviewing Policy:**

Affirmed.

**Final Justification:**

The rebuttal fully resolves my concerns; I increased my overall score.

**Key Questions For Authors:**

Can you give more details on the baseline approaches and the benchmarks?

**Limitations:**

yes

**Strengths And Weaknesses:**

The paper appears to be well-presented and sound.

The paper presents as main contribution the first general-purpose multi-agent framework for abductive reasoning. So, the paper is certainly original. It uses a neuro-symbolic design with causal graphs and a state machine to build explicit belief states, turning open-ended reasoning into a more directed and convergent search process. Experiments on two real-world datasets show that the approach outperforms existing baselines. However, the baseline approaches and benchmarks a scarcely described. So, I'm not yet convinced about the significance of the work.

---

> ### Author Rebuttal · Authors · 2026-03-25
>
> We thank the reviewer for the careful review and for recognizing the soundness and presentation of our paper. In the following, we provide more details on the baseline approaches and benchmarks to better support the significance of the work.
>
> ---
> **W1 & Q1: More details on the baseline approaches and the benchmarks.**
>
> **A1:**
> - **Scope of details already provided in the paper:** The construction of the baselines, parameter settings, role design for Multi-Agent baselines, benchmark construction, task workflows, data reliability controls, dataset descriptions, and evaluation protocols are already described in the paper (Sec. 4, Sec. 4.1, Sec. 4.2, and Appendix C).
> - **How the 8 baselines are instantiated and executed:** The 8 baselines are formed by the Cartesian product of two dimensions: agent architecture (Single-Agent vs. Multi-Agent) and reasoning topology (CoT, ToT, GoT, FoT). More concretely, in the Single-Agent setting, CoT follows a linear ReAct trajectory, where the agent iteratively performs thought, action (e.g., check the metric of cpu_util, execute the shell ```free -h```), and observation once received the input query, and advances the reasoning process in a chain structure to reach the answer. ToT extends this process into a tree structure, allowing the agent to explore multiple candidate reasoning paths (i.e., alternative hypotheses or lines of thought), continue search along promising branches, and backtrack when necessary. GoT and FoT organize reasoning nodes as graph and forest structures, respectively, to support richer exploration. In the Multi-Agent setting, the same reasoning topologies are preserved, but the process is coordinated by a central agent that performs global planning and control, dispatches role-specialized expert agents to execute domain-specific actions, and then integrates their returned evidence and analyses for the next reasoning step. Taken together, these baselines cover the major ways current agent-based reasoning systems organize reasoning trajectories and agent collaboration.
> - **Details on two benchmarks:** Both benchmarks are designed to reflect complex real-world abductive tasks rather than knowledge-recall problems. In medical diagnosis, we use real pathological cases from DiagnosisArena, where each case involves a non-trivial diagnostic scenario requiring careful examination and evidence-based analysis, rather than solving basic medical multiple-choice questions such as MedQA. In failure diagnosis in distributed systems, the cases are also drawn from complex production incidents, e.g., faults such as abnormally high MySQL InnoDB buffer pool utilization, which require engineers to iteratively inspect heterogeneous evidence before reaching the root cause. These two benchmarks therefore reflect the kinds of abductive tasks that physicians and on-call engineers must spend substantial effort solving in real practice. More detailed dataset descriptions are provided in our anonymous repository https://anonymous.4open.science/r/Graph-of-States-5B4E. For the failure diagnosis benchmark, due to data sensitivity, the dataset is not yet publicly released; we are currently conducting data desensitization and security/compliance review, and plan to release it once these procedures are completed.
>
> We hope this clarification addresses your concern, and if you find our response helpful, we kindly request the reviewer to consider adjusting the rating.

---

> > ### Author Rebuttal · Reviewer_oMNi · 2026-04-04
> >
> > Fully resolved.

---

> > > ### Author Response · Authors · 2026-04-04
> > >
> > > Thank you for your thoughtful and insightful comments. We sincerely appreciate your time and effort in reviewing our manuscript. We are pleased that our responses have addressed your concerns, and we will further refine the manuscript according to your valuable suggestions.

---

### Decision · Program_Chairs · 2026-04-30

**Decision:**

Accept (regular)

**Comment:**

This paper proposes Graph of States (GoS), a neuro-symbolic framework that uses a causal graph and state machine to support explicit belief-state management for abductive reasoning with LLMs, demonstrating substantial improvements over eight baselines on two real-world benchmarks. Reviewers acknowledged the practical significance and strong empirical results, while raising concerns about the deduction/abduction framing, limited neuro-symbolic baseline coverage, and under-specified confidence mechanism; these were largely addressed in the rebuttal, with three of four reviewers recommending acceptance.